



# Different coastal marsh sites reflect similar topographic conditions for bare patches and vegetation recovery

Chen Wang[1,2], Lennert Schepers[2], Matthew L. Kirwan[3], Enrica Belluco[4], Andrea D'Alpaos[5], Qiao Wang[6,1], Shoujing Yin[1], and Stijn Temmerman[2]

[1]Satellite Application Center for Ecology and Environment, Ministry of Ecology and Environment / State Environmental Protection Key Laboratory of Satellite Remote Sensing, Fengde East Road 4, Beijing 100094, China
[2]Ecosystem Management Research Group, University of Antwerp, Universiteitsplein 1, 2610 Wilrijk, Belgium
[3]Virginia Institute of Marine Science, PO Box 1346, 1375 Greate Road, Gloucester Point, Virginia 23062, USA
[4]Department of Civil, Environmental, and Architectural Engineering, University of Padova, via Loredan 20, 35131 Padua,
Italy
[5]Department of Geosciences, University of Padua, Via Gradenigo 6, 35131 Padua, Italy
[6]Faculty of Geographical Science, Beijing Normal University, Xinjiekouwai Street 19, Beijing 100875, P.R.China

*Correspondence to*: Qiao Wang (wangqiao@mee.gov.cn) and Shoujing Yin (shoujingy@163.com)

**Abstract.** The presence of bare patches within otherwise vegetated coastal marshes is sometimes considered to be a
symptom of marsh die-back and the subsequent loss of important ecosystem services. Here we studied the topographical conditions determining the presence and revegetation of bare patches in three marsh sites with contrasting tidal range, sediment supply and plant species: the Scheldt Estuary (the Netherlands), Venice Lagoon (Italy), and Blackwater Marshes (Maryland, USA). We analyzed topographic properties of bare patches, including elevation, size, distance and connectivity to channels based on GIS analyses of aerial and LIDAR imagery. Our results demonstrate that across the different marsh
sites, bare patches connected to channels occur most frequently at the lowest elevations and farthest distance from the main channels. Bare patches disconnected from channels occur most frequently at intermediate elevations and distances from channels, and vegetated marshes dominate at highest elevations and shortest distances from channels. Revegetation in bare patches is observed in only one site with the highest tidal range and highest sediment availability, and preferentially occurs from the edges of small unconnected bare patches at intermediate elevations and intermediate distances from channels. Our
results are discussed within the alternative stable state theory. We suggest the existence of two stable states, a high-elevated vegetated state close to channels that tends to remain high and vegetated, and a low-elevated state of bare connected patches far from channels that tends to remain bare, with an unstable state at intermediate channel distances where bare patches may form and rapidly become revegetated.

## 1 Introduction

Tidal marshes are coastal ecosystems that provide many valuable ecosystem services such as fishery production (Barbier et al., 2011), sequestration of $CO_2$ (McLeod et al., 2011), protection against shoreline erosion and mitigation of flood risks during storm surges (Barbier et al., 2008; Wamsley et al., 2010; Gedan et al., 2011; Temmerman et al., 2013; Temmerman





and Kirwan, 2015). However, tidal marshes and their valuable ecosystem services can be lost when marshes die-off, for instance, as a consequence of sea level rise. Large-scale tidal marsh loss by conversion of marshes into bare tidal flats, open water or bare patches within marshes has been reported from different locations around the world (Baumann et al., 1984; Day et al., 2000; Kearney et al., 2002; Carniello et al., 2009; Kirwan and Megonigal, 2013). Bare patches within marshes, which are often covered by standing water and then referred to as pools, ponds (Stevenson et al., 1985) or marsh basins (Mariotti and Fagherazzi, 2013), are a common feature in salt marshes around the world. In some regions, bare patches are dynamic features that develop but also recover and revegetate (e.g., New England; Wilson et al., 2009, 2010, 2014). In other areas, however, bare patches do not revegetate and are causing permanent marsh loss on a large scale (e.g. Mississippi Delta Penland et al., 2000; Morton et al., 2003).

Marsh loss and recovery is of particular concern because there is growing evidence that vegetated marshes and bare flats behave as alternative stable ecosystem states (Fagherazzi et al., 2006; Kirwan and Murray, 2007; Marani et al., 2007; van Wesenbeeck et al., 2008; Marani et al., 2010; D'Alpaos, 2011; McGlathery et al., 2013; Wang and Temmerman, 2013; Moffett et al., 2015; D'Alpaos and Marani, 2016; van Belzen et al., 2017), which implies that recovery after marsh loss would be very difficult (Hu et al., 2015a; van Belzen et al., 2017). Observations have shown that vegetated marshes and bare flats occupy different elevation ranges (Marani et al., 2007, 2010; Carniello et al., 2009; Wang and Temmerman, 2013) and that shifts from the low-lying bare state to the high-elevated vegetated state occur rapidly once a threshold elevation has been exceeded (Wang and Temmerman, 2013). Moreover, models indicate that the system would shift abruptly between the high elevation vegetated state and low-lying bare state when a threshold value is reached in elevation, sediment input, or rate of sea level rise (Fagherazzi et al., 2007; Kirwan and Guntenspergen, 2010; Marani et al., 2010; D'Alpaos et al., 2011; D'Alpaos and Marani, 2016). Previous studies further suggest that the state shift can be irreversible because of a hysteresis effect (Kirwan and Murray, 2007; Marani et al., 2010; Kirwan et al., 2011), where the threshold conditions to revert the ecosystem back to the original state are far more difficult to reach than the threshold conditions that caused the shift (Scheffer et al., 2001; Scheffer and Carpenter, 2003). Field experiments have also demonstrated that vegetation recovery after disturbance is slower under increased tidal inundation, which further suggests the applicability of alternative stable state theory to vegetated and bare areas in intertidal zones (van Belzen et al., 2017).

The two stable states of marshes and tidal flats can be explained by positive feedback mechanisms which are strongly mediated by the presence or absence of marsh vegetation. As long as vegetation is present on the marsh, waves and tidal currents are effectively attenuated by vegetation-induced friction over several meters of continuously vegetated marsh surfaces (Neumeier and Amos, 2006; Mudd et al., 2010; Vandenbruwaene et al., 2011; Yang et al., 2012; Hu et al., 2014). As a consequence, suspended sediment is deposited on the marsh surface and marshes can maintain a high position in the tidal frame, even with sea level rise (Kirwan and Guntenspergen, 2010; D'Alpaos et al., 2011; Fagherazzi et al., 2012). Above- and belowground plant material further contributes to marsh accretion (Nyman et al., 2006; Kirwan and Guntenspergen, 2012). When vegetation is absent, however, organic matter accumulation is strongly reduced, and increased tidal currents and waves may prevent sedimentation or even trigger erosion (Fagherazzi et al., 2006; Kirwan and Murray,



2007; Marani et al., 2007; Mariotti and Fagherazzi, 2010; Temmerman et al., 2012). In large lagoons or extensive tidal basins, the low elevation of the tidal flats is mainly maintained by wave erosion (Fagherazzi et al., 2006; Mariotti and Fagherazzi, 2010; Hu et al., 2015b). The existence of these two alternative stable states has been empirically observed on the

large scale of whole tidal basins where large areas (~km ꝫ) of marshes and tidal flats may coexist next to each other (Marani et al., 2007; Carniello et al., 2009; Wang and Temmerman, 2013). However, the existence of alternative stable states has not yet been empirically explored to explain marsh loss and recovery by formation and revegetation of bare patches (~10–100 m ꝫ, which is addressed in this paper.

Bare patches are defined here as non-vegetated areas in the interior of otherwise vegetated marshes. Here we consider two

types of bare patches: (i) connected bare patches that have a connection to the tidal channel network and (ii) isolated bare patches that are separated from the channels by surrounding marsh vegetation. Literature suggests that unconnected bare patches start as areas with vegetation die-off, by increased flooding stress and inadequate drainage, high salinity stress (DeLaune et al., 1994; Wilson et al., 2009, 2014), coverage by drifted plant material (Harshberger, 1916; Miller and Egler, 1950; Redfield, 1972), physical disturbance by ice, or herbivory by crabs, nutria, muskrats, geese or snails (Harshberger,

1916; Stevenson et al., 1985; DeLaune et al., 1994; Silliman, 2005; Argow and FitzGerald, 2006). Subsequent elevation loss due to the collapse of the root structure or decomposition and disintegration of soil organic matter can deepen the bare patches (DeLaune et al., 1994; Wilson et al., 2014). Connected bare patches form by creek-bank erosion at the creek heads (Kearney et al., 1988) and subsequent connection of channel heads to bare patches (Redfield, 1972) or by expansion of unconnected bare patches that ultimately reach a channel and become hydraulically connected to the channel network

(Wilson et al., 2014; Mariotti, 2016).

Nevertheless, it is not fully understood under which topographic conditions connected and unconnected bare patches occur, and especially under which conditions they recover through re-establishment of vegetation. For example, we may hypothesize that unconnected bare patches are buffered from tidal currents and waves by the surrounding marsh vegetation, and therefore are less prone to erosion and more suitable for vegetation recovery. On the other hand, they might also receive

less sediment input since sediment is efficiently trapped by the surrounding vegetation buffer (Mudd et al., 2010; Moskalski and Sommerfield, 2012). The opposite applies for connected bare patches: they receive direct sediment input through the channels, but experience higher flow velocities that may cause sediment resuspension and erosion. Some studies show that marsh plants might recolonize bare patches when they become connected, drain and if vertical accretion elevates the bare patches sufficiently for plant establishment (Redfield, 1972; Wilson et al., 2009, 2014). However, higher flow velocities and

therefore a decrease in accretion by reduced mineral sediment deposition or erosion may inhibit the recovery of vegetation in connected bare patches (DeLaune et al., 1994; Mariotti, 2016).

Hence, despite the fact that bare patches are often recognized as symptoms of marsh loss (Kearney et al., 1988; DeLaune et al., 1994; Fagherazzi et al., 2013; Mariotti and Fagherazzi, 2013; Wilson et al., 2014; Mariotti, 2016), there are relatively few studies on the dynamics of bare patches. For example, the modelling study by Mariotti (2016) simulates that pond

expansion is favored under conditions with low tidal range, low sediment supply and high relative sea level rise. Apart from





this study, there is poor empirical evidence on the conditions that determine the presence and/or recovery of bare patches, especially across marsh sites that differ in characteristics such as tidal range, sediment supply and plant species. In this paper, we first study the topographic conditions determining the presence of bare patches. Next, we study the topographic conditions determining the marsh vegetation recovery (i.e., the re-establishment of vegetation) in bare patches. To identify

the topographic conditions determining the presence of bare patches, we compared the surface elevation, bare patch size and distance to channels for connected and unconnected bare patches in three different sites, located in the Scheldt Estuary (a river-dominated estuary in the Netherlands, 4.8 m tidal range), Venice lagoon (a back-barrier lagoon in Italy, 1.0 m tidal range) and Blackwater Marshes (a submerging tidal marsh, in Maryland, USA, < 0.5 m tidal range). To identify the conditions determining the revegetation of bare patches, we carried out a time series analysis in the Scheldt Estuary, the only

site where revegetation was observed and searched for relations between the rate of revegetation of bare patches and topographic conditions including surface elevation, distance to channels and the width of connecting channels. Our hypotheses are that (1) bare patches across all three study sites are found at similar elevation relative to the tidal frame, distance to tidal channels and degree of connectivity to tidal channels; (2) low elevation relative to the tidal frame and wide channel connection lead to larger bare patches that are more difficult to revegetate.

## 115 2 Study area

We studied three marsh sites that have different characteristics, including different tidal range, sediment supply and plant species: (i) Saeftinghe (the Netherlands), (ii) San Felice (Italy) and (iii) the Blackwater Marshes (USA). Within each of the three marsh sites, a specific study area was selected based on data availability and the presence of bare patches on aerial images. In the next paragraph we give more background information on the three marsh sites. Detailed information about the

aerial images is provided in Sect. 3.

### 2.1 Saeftinghe marsh, Scheldt estuary, the Netherlands

The Scheldt estuary is a river-dominated estuary located in the southwest of the Netherlands and the northwest of Belgium (Fig. 1). The Saeftinghe marsh (51.33° N, 4.17° E) is a 3000 ha tidal marsh within the brackish zone of the estuary. It is subject to a semi-diurnal tidal regime with a local mean tidal range of 4.88 m, a salinity of 5–18 PSU, and a mean suspended

sediment concentration (SSC) of 30–60 mg L$^{-1}$ (Temmerman et al., 2003a; van Damme et al., 2005). In the last 80 years, a long-term rise of mean high water level (MHWL) was observed in the Saeftinghe marsh at a rate of 5.7 mm/yr, while the vegetated marsh regions expanded in area and increased in elevation steadily and continuously (Wang and Temmerman, 2013). The lower areas are colonized by the pioneer plant species *Spartina anglica* and *Salicornia europaea*. *Scirpus maritimus* is found in depressions of higher marshes. *Elymus athericus* is present on natural levees along creek edges. The

highest parts are dominated by *Phragmites australis*. Marsh vegetation is observed between -2 m and +1 m relative to MHWL, with the highest frequency of vegetation presence centered around MHWL (Wang and Temmerman, 2013). Parts of





the Saeftinghe marsh have been converted to bare patches. This is partly attributed to geese grubbing for below-ground tubers (Elschot et al., 2017). In addition, bare patches are formed at places with poor drainage and temporary ponding of water after high overmarsh tides. This is especially the case at marsh platforms near the head of the smallest tidal channels,
i.e. where drainage towards channels is least developed. This is the case in the selected study site, situated in the south of the Saeftinghe marsh, covering an area of 35 ha (Fig. 1).

## 2.2 San Felice marsh, Venice lagoon, Italy

The Venice lagoon is a back-barrier tidal lagoon situated in the northeast of Italy and is characterized by a micro-tidal semi-diurnal regime with a mean tidal range of about 1.0 m (Day et al., 1999) and a maximum tidal range of 1.5 m (Rinaldo et al.,
1999a, 1999b; Marani et al., 2007). The long-term rate of relative sea level rise varies around 3–4 mm/yr (Carbognin et al., 2004). The marsh systems in the Venice lagoon are degrading with about 75 % reduction in marsh area since 1901 (from a 64 km$^2$ to 43 km$^2$), caused by both drowning and lateral erosion of marshes (Tommasini et al., 2019). The San Felice salt marsh (45.48˚N, 12.46˚E) is located in the northern part of the Venice Lagoon, close to the Lido inlet (see Marani et al., 2003 for further details on the study site) and is considered to be one of the best preserved marshes in the Venice Lagoon, being
capable of keeping pace with current relative sea level rise (e.g., Roner et al., 2016). The average salinity varies between 24 and 33 PSU (Gieskes et al., 2013; Zirino et al., 2014), and the average SSC is between 10 and 20 mg/l (Zaggia and Ferla, 2005; Defendi et al., 2010; Venier et al., 2014). The salt marsh is occupied by halophytic species (Silvestri et al., 2005; Marani et al., 2006). The pioneer species present on the lowest elevations are mainly *Salicornia veneta* and *Spartina maritima. Limonium narbonense* covers slightly higher elevations. *Sarcocornia fruticosa* dominates the highest elevations,
such as natural levees, together with *Puccinellia palustris* and *Inula crithmoides*. *Juncus maritimus* is observed within a broad range of elevations (Silvestri et al., 2005). The elevation of the salt marsh ranges from 0 m to 0.7 m relative to mean sea level (MSL). Our specific study site in the interior of the San Felice marsh has an area of 72.3 ha (Fig. 2).

## 2.3 Blackwater marshes, Chesapeake Bay, USA

The Blackwater marshes (38.40˚ N, 76.08˚ W), part of the largest marshland in the Chesapeake Bay, are situated at the
confluence of the Blackwater and Little Blackwater Rivers. They cover an area of about 6000 ha with an average SSC of about 50 mg/L and an average salinity of 10 PSU (Stevenson et al., 1985; Ganju et al., 2013; Kirwan and Guntenspergen, 2015). Long-term local sea level rise is currently 3.7 mm/yr (NOAA station 8571892, http://tide-sandcurrents.noaa.gov/sltrends, 12/19/2016). Extensive marsh loss was reported in the Blackwater system, where about half of the interior marshes have disappeared since 1938, mainly by the development and enlargement of bare patches, which are
occurring as interior marsh pools (Stevenson et al., 1985; Kearney et al., 1988; Kirwan and Guntenspergen, 2012; Schepers et al., 2017). The marsh loss and pool expansion has been attributed to submergence by sea level rise and vegetation disturbance by invasive herbivores and subsequent open-water expansion (Stevenson et al., 1985; Kendrot, 2011). Changes in water level are mainly driven by wind setup and barometric pressure fluctuations during meteorological events, while the





astronomical tidal range is about 0.25 m at our study site. Brackish vegetation dominates the Blackwater marshes, with
species such as *Scirpus americanus* and *Spartina alterniflora* occupying low elevations, and *Spartina patens*, *Distichlis spicata, Spartina cynosuroides,* and *Phragmites australis* occupying higher elevations (Pendleton and Stevenson, 1983; Kirwan and Guntenspergen, 2012). Our specific study area in the Blackwater Marshes covers an area of about 699.8 ha (Fig. 3).

## 3 Materials and data preprocessing

### 3.1 General procedure

For all three study sites, aerial photographs were digitized, georeferenced and manually classified into vegetated marshes, unconnected bare patches, connected bare patches and tidal channels (Figs. 1-3). Bare patches that were smaller than 1 m$^2$ were not considered in this study. Given the resolution of the images (see below), bare patches were classified as to be connected to the channel network when the connecting channel was at least 0.5 m wide. Hence our classification of
unconnected bare patches includes truly unconnected patches, but may also include patches with a small connecting channel (less than 0.5 m wide) that was impossible to detect on the aerial images. LIDAR data (Figs. S1-S3) was used to analyze the elevation differences between vegetated marshes, unconnected bare patches, connected bare patches and tidal channels. When bare patches were inundated during the LIDAR survey, the soil surface elevation within the bare patches was measured with field surveys (methods are explained below for the different study sites). Generally, LIDAR data have larger
and more homogeneous spatial coverage and higher spatial resolution. Field surveys only include selected locations, but with greater vertical accuracy, especially for vegetated areas where LIDAR partially reflects on the vegetation canopy, and open water where LIDAR reflects on the water surface. All the spatial analyses were done using ArcGIS.

### 3.2 Saeftinghe

For the Saeftinghe study site, a time series of false-color aerial images was used, from 1990, 1998, 2004 and 2008. The four
images were selected considering the data availability and to detect dynamic changes from vegetated marsh portions into bare patches and vice versa. All the photos were processed in a similar way, by scanning, georeferencing and mosaicking them into digital pictures with a minimum resolution of 0.5 m. All the aerial images were provided by Rijkswaterstaat (the Dutch governmental institute for water management) (Huijs, 1995; van der Pluijm and de Jong, 1998; Reitsma, 2006; Bakker and Bijkerk, 2009). From all the available aerial photographs, we extracted two sample areas (Fig. 1) free from drifted plant
debris, which were analyzed together. The digitized aerial images in the sample areas were classified into vegetation, water and bare soil based on supervised maximum likelihood classification, and then further classified visually into vegetated marshes, channels, connected bare patches and unconnected bare patches. For elevation data in Saeftinghe, we used a Digital Terrain Model (DTM) with a resolution of 2×2 m (Fig. S1), which was obtained from a LIDAR survey performed in 2004 during low tide with a maximum vertical error of 0.2 m (Alkemade, 2004). The measurement point density of the LIDAR



survey varied from 1 point per 16 m$^2$ to several points per m$^2$. The DTM data were also provided by Rijkswaterstaat. We used only one LIDAR dataset to derive the elevations of bare patches and marshes over the period 1990-2008 because previous research in the area showed that during that period, elevation changes were limited with maximum rates of 1 cm/yr (Wang and Temmerman, 2013). This implies that over the considered time scale (1990-2008), maximum elevation changes (~18 cm in 18 years) are of the same order of magnitude as the vertical error of the LIDAR data (~20 cm). Therefore, we

decided to use one LIDAR-based DTM for 2004, which is considered to be representative to characterize the approximate time-averaged elevation of marshes and bare patches over the period 1990-2008.  No field survey data were used for Saeftinghe since all bare patches drain completely during neap tides so that soil surface elevations were recorded by LIDAR. We note that even so-called unconnected bare patches may have drainage via connecting channels < 0.5 m wide (see above) or truly unconnected bare patches may drain at neap tides by subsurface drainage towards nearby channels, that are typically

1-3 m deep and also dry during low tides in this macro-tidal setting (mean tidal range of 4.9 m).

### 3.3 San Felice

For the San Felice study site, our analysis was based on a vegetation map classified from a hyperspectral image with a resolution of 1.3 m, which was acquired in 2002 by the airborne CASI sensor (15 bands in the visible and near infrared portion of the spectrum) (Belluco et al., 2006). The vegetation map distinguished water, bare soil and four vegetation classes.

It was visually reclassified into channels, connected bare patches, unconnected bare patches and vegetated marshes (Fig. 2). For the latter, we consulted a black-white aerial photograph acquired in 2000 with a resolution of 16 cm, and a 1-meter resolution pan-sharpened multispectral IKONOS image acquired in 2006. For elevation data in San Felice, we used both a DTM obtained from a LIDAR survey (Fig. S2) and field measurements. The LIDAR survey was performed during low tide in 2002 with a mean measurement point density of about 48 points/m$^2$ and a vertical accuracy better than 0.15 m (Wang et

al., 2009). From these data we constructed a gridded DTM with a spatial resolution of 1×1 m. Field elevation measurements from the Venice Water Authority in 2000 were also used, because some bare patches were inundated during the LIDAR survey. Data were collected with stereo aerial photography for marshes, stadia rods with GPS for areas close to marshes and mudflats, and single-beam echo-sounder for shallow waters (Sarretta et al., 2010). In total, 340 elevation measurements were located in vegetated marshes, and 95 measurements in bare patches. The boundary of the study area was delineated by

channels and creeks as shown in Fig. 2, considering the availability of data. Since almost no vegetation recovery in bare patches was observable on aerial images from the San Felice marsh, we did not do a time series analysis on vegetation recovery.

### 3.4 Blackwater

In the Blackwater study site, we selected a study area away from the influence of roads and uplands (Fig. 3). The small study

area (marked with shading in Fig. 3) was chosen for the field survey. A larger study area (the entire colored region in Fig. 3) was later considered in order to increase the number of bare patches connected to channels wider than 1 m. Bare patches that





are connected with narrow channels (< 1 m) and that are located outside of the small study area (blue polygons in Fig. 3) were not considered in the analysis. We used false color aerial photographs with a spatial resolution of 0.3 m obtained in 2010 and provided as digitized and georeferenced mosaic by the United States Department of Agriculture (USDA). Similar

as in Saeftinghe, we classified the photos into water, vegetation and bare soil using a supervised maximum likelihood classification procedure, and then we visually classified them into vegetated marshes, connected bare patches, unconnected bare patches and channels. We also used data acquired from a LIDAR survey (Fig. S3) and a field elevation survey. The LIDAR data were obtained in 2003 with an average area sampling density of about 0.8 points per m$^2$ and a mean vertical accuracy of 0.14 m. The DTM was provided with a resolution of 2×2 m by the U.S. Geological Survey and Maryland

Department of Natural Resources. As most bare patches were covered by water during the LIDAR survey, a field survey was carried out in 2012 in the small study area using RTK-GPS with ±1.5 cm accuracy. In total, 36 elevation measurements were collected in 5 unconnected bare patches, 31 measurements in 5 connected bare patches and 93 measurements in the vegetated marshes. An overview of the number of data points (LIDAR and GPS measurements) that fall within marshes and bare patches are given for the different study sites in Table 1. Other studies in the Blackwater Marshes have demonstrated that

recovery of marsh vegetation within bare patches is absent (Schepers et al., 2017), and therefore we did not analyze a time series of aerial images.

## 4 Data analysis

### 4.1 Topographic conditions determining the presence of bare patches

In order to identify the topographic conditions determining the presence of bare patches or marsh vegetation, we analyzed

the frequency distributions of surface elevation and distance to channels for connected and unconnected bare patches, and compared them with the vegetated marsh portions, for the three study sites. The surface elevation was analyzed using LIDAR data and field data. The distance to channels was calculated as the Euclidean Distance from the edge of channel polygons. Bare patches smaller than 1 m$^2$ were excluded from the analysis. The nonparametric Mann-Whitney U test was conducted to test whether or not there were significant differences between elevations of vegetated marsh portions,

connected and unconnected bare patches.

Elevation classes of 10 cm were used, since smaller elevation classes were not deemed to be reasonable considering the vertical accuracy of LIDAR data. Surface elevation relative to the local mean low and high water levels (i.e., the tidal frame), is an important factor for vegetation because it determines the frequency, depth and duration of tidal flooding and is widely considered as a crucial ecological condition for marsh plant growth. Therefore, in order to allow comparisons between the

three marsh sites with largely different tidal ranges, we rescaled the surface elevation relative to the tidal frame using the following relationship:

$$RE = \frac{E - MLWL}{MHWL - MLWL} \tag{1}$$





where *RE* is the relative elevation (a dimensionless proportion of the local tidal frame), *E* is the actual elevation (in m relative to a fixed datum), *MLWL* and *MHWL* are the mean low water level and mean high water level, respectively (in m

relative to the same datum). Hence *RE* is 0 for elevations equal to *MLWL* and is 1 for elevations equal to *MHWL*.

In addition, the frequency distribution of bare patch sizes was calculated and related to the widths of channels that were connected to bare patches. The channel width was measured on the aerial photographs at the connection with the bare patch for each single patch and classified into categories with 5 m spacing. Unconnected bare patches (channel width < 0.5 m) and bare patches connected with small channels (channel width between 0.5 m and 1 m) were classified as two separate

categories because of their large number. We combined all bare patches with a connection > 80m in the highest class, since there were only 0, 1 and 2 patches for this category in the Saeftinghe, San Felice, and Blackwater marsh sites, respectively.

## 4.2 Topographic conditions determining the revegetation of bare patches in Saeftinghe

We studied revegetation of bare patches in the Saeftinghe marsh during the last two decades. We did not include the San Felice and Blackwater marshes in this analysis, because there was almost no revegetation recognizable on the aerial

photographs during this period of the last two decades. This does not necessarily mean that revegetation of bare patches is not taking place in the latter two study areas, but at least suggests that it is not occurring on the time scale of the last two decades. Between each aerial photograph in 1990, 1998, 2004 and 2008, we identified areas that changed from vegetated to bare surfaces, areas that revegetated from bare to vegetation, and areas that remained bare or vegetated. From this data, we determined the rate of revegetation of bare areas. We made a distinction between the following classes:

275        (1) permanent bare patches that never revegetated within the considered time period from 1990 to 2008;

(2) rapidly revegetated bare patches, identified as bare in only one image, either 1998 or 2004, and observed as vegetation in the other three images;

(3) permanent marsh areas, classified as vegetation throughout the time series.

In order to identify the topographic conditions for rapid or no revegetation of bare areas, the frequency distribution of

elevation was calculated for these three classes (permanent bare patches, rapidly revegetated bare patches, and permanent marsh areas), as well as the frequency distributions of the distance to the closest channel. In addition, we also determined the width of the channels connecting to the bare patches. For permanent bare patches, the channel width is calculated as the mean value for 1990, 1998, 2004 and 2008. For rapidly revegetated bare patches, the channel width is the value when the bare patches occurred, either in 1998 or in 2004. In order to identify the relationship between the rate of revegetation and the

width of connecting channels, the frequency distribution of channel widths was compared between permanent bare patches and rapidly revegetated bare patches.





## 5 Results

### 5.1 Topographic conditions determining the presence of bare patches

In order to identify the topographic conditions determining the presence of bare patches, we tested relationships between their presence and three topographic variables, which are (1) elevation of the bare soil surface, (2) distance of the bare patches from channels and (3) channel width for bare patches connected to channels (channel width <0.5 m for unconnected bare patches). We first tested whether these three topographic variables are independent from each other. The correlations were low (Pearson's r <0.5) and not significant (p>0.05) between all variables and for all field sites. Only for the Blackwater marsh, the correlation between the elevation and the channel width was high (Pearson's r = -0.9), but this correlation is based

on a very low number of connected (n=5) and unconnected bare patches (n=5).

### 5.1.1 Elevation

In Saeftinghe, the connected bare patches, unconnected bare patches and vegetated marshes fall within the elevation ranges of 2.3–3.5 m above MSL, which is close to the local MHWL (relative elevation $RE$ = 0.91–1.16) (Fig. 4a). The differences in elevation between the vegetated marshes, connected and unconnected bare patches were statistically significant between

each two of the three features (p < 0.001 based on the Mann-Whitney test). The peaks of the elevation distribution for the vegetated marshes and unconnected bare patches are 0.1 m higher than for the connected bare patches (or the difference between relative $RE$, $\Delta RE$ = 0.02). The mean elevation of the vegetated marshes is highest (2.97 m above MSL, $RE$ = 1.05), whereas this is 0.14 m lower for the unconnected bare patches ($\Delta RE$ = 0.03) and 0.23 m lower for the connected bare patches ($\Delta RE$ = 0.05).

In San Felice, the connected bare patches, unconnected bare patches and vegetated marshes are situated in different ranges of elevations between -0.5 and +0.7 m relative to MSL ($RE$ = 0–1.2, Fig. 4b). The differences in elevation distributions of these three categories are also statistically significant (p < 0.001 based on the Mann-Whitney test). The elevation measured in the field is lower than that from the LIDAR survey for both the connected bare patches and vegetated marshes. The peaks of the elevation distribution of the vegetated marshes and unconnected bare patches are about 0.15 m lower than MHWL ($RE$ =

0.85) based on LIDAR data, and about 0.3 m or 0.5 m higher than connected bare patches ($\Delta RE$ = 0.3 or 0.5) based on LIDAR or field data, respectively. The mean LIDAR elevation of the vegetated marshes is 0.35 m relative to MSL ($RE$ = 0.85), which is 0.04 m higher than unconnected bare patches ($\Delta RE$ = 0.04) and 0.28 m higher than connected bare patches ($\Delta RE$ = 0.28).

In the Blackwater Marshes, connected bare patches, unconnected bare patches and vegetated marshes occupy significantly

different ranges of elevations (p < 0.001 based on Mann-Whitney test) between -0.7 m and +0.5 m relative to MSL ($RE$ = -0.9–1.5, Fig. 4c). The peaks of the elevation distribution of the vegetated marshes are 0.1 m lower than MHWL ($RE$ = 0.8), 0.3 m higher than unconnected bare patches ($\Delta RE$ = 0.6) and 0.6 m higher than connected bare patches ($\Delta RE$ = 1.2). The





mean elevation is the highest for the vegetated marshes (0.13 m relative to MSL, *RE* = 0.76), 0.23 m lower for the unconnected bare patches (*ΔRE* = 0.46) and 0.6 m lower for connected bare patches (*ΔRE* = 1.2).

Together these results indicate that connected bare patches, unconnected bare patches, and vegetated marshes tend to occupy different elevation ranges at each site (p < 0.001 by Mann-Whitney), with the largest absolute elevation differences in Blackwater, the smallest in Saeftinghe, and intermediate values for San Felice. Connected bare patches always lie within the lowest elevation range, whereas vegetated marshes always dominate the highest elevation range around MHWL. Unconnected bare patches are always found in the intermediate elevation range, which is about 0.1–0.5 m higher than the

connected bare patches. The difference in *RE* (relative to the tidal frame) between the connected and unconnected bare patches is about 0.02 in Saeftinghe, 0.2–0.5 in San Felice, and 0.6–0.8 in Blackwater.

### 5.1.2 Distance to channels

The frequency distribution of the distance between a bare patch and closest channel shows similar results for the three marsh sites (Fig. 5). Vegetated marshes rather than bare patches occur near channels. With increasing distance from channels,

marsh vegetation becomes less frequent and unconnected bare patches become more frequent. Connected bare patches occur most frequently at large distances from the channels. The peak of the distribution is situated at 1.0 m for vegetated marshes in all three sites; at 8 m for unconnected bare patches and over 10 m for connected bare patches in both Saeftinghe and San Felice; and at 82 m and 89 m for unconnected and connected bare patches in Blackwater, respectively.

### 5.1.3 Bare patch size in relation to connectivity to channels

Bare patch size generally increases with increasing width of connecting channels, whereas the number of bare patches decreases with increasing channel widths (Fig. 6). The unconnected bare patches in Saeftinghe, San Felice and the small study area of Blackwater, occupy 63 %, 36 % and 67 % of the total number of bare patches, respectively, but only 2 %, 1 % and 3 % of the total area of bare patches, respectively. Hence, unconnected bare patches are numerous but small. The number of connected bare patches, in contrast, is in most cases smaller and they become less abundant with increasing width

of the connecting channels.

### 5.2 Topographic conditions determining the revegetation of bare patches in Saeftinghe

The multi-temporal analysis for Saeftinghe shows that bare patches have been dynamically expanding or shrinking between the four images of 1990-1998-2004-2008 (Fig. 7). We focused on bare areas with two extreme rates of revegetation, which are permanent bare areas (which never revegetated throughout the time series) and rapidly revegetated bare areas (only

present in 1998 or 2004 and revegetated by the next time step). The spatial distribution of these bare categories (Fig. 7) suggests that the inner portion of big connected bare patches tends to be stable and never revegetated within the studied period, while rapidly recovering bare areas are mainly present at the edge of small bare patches.



### 5.2.1 Elevation

The elevation distribution showed that permanently bare areas (i.e. remaining bare over the studied 18-year period) occupy
the lowest range of elevations, whereas permanent marsh areas have the highest range of elevations (Fig. 8a). At intermediate elevations, bare patches become rapidly revegetated (i.e. within 4 to 6 years after their first appearance (Fig. 8a).

### 5.2.2 Distance to channels

The frequency distribution of the different bare categories with distance to the channels (Fig. 8b) shows that stable marshes are closest to channels with a peak around 1-2 m from channels. Bare areas that revegetated quickly have an intermediate
distance around 8 m from channels, whereas permanent bare areas are located farthest from the channels with a peak at 21 m.

### 5.2.3 Connectivity to channels

Permanent bare areas are always connected to channels, and tend to be associated with wide channels (Fig. 8c). The percentage of bare areas that become revegetated increases with decreasing channel width (Fig. 8c).

### 6 Discussion

Bare patches within otherwise vegetated coastal marshes are often recognized as symptoms of marsh loss in many places around the world (Kearney et al., 1988; Fagherazzi, 2013; Mariotti and Fagherazzi, 2013; Ortiz et al., 2017; Schepers et al., 2017), but comparative studies among different marsh systems to better understand the conditions that determine their presence and potential vegetation recovery are relatively scarce (e.g., Mariotti, 2016). Fig. 9 provides a schematic summary of our results, and of our interpretations that are discussed here. For three marsh sites with different tidal ranges, sediment
input and plant species, we showed that: (1) bare patches connected to channels occur most frequently at the lowest surface elevations and farthest distances from creeks; unconnected bare patches most frequently occupy intermediate elevations and distances from creeks, and are smaller in size and larger in number; and vegetated marshes dominate at the highest surface elevations and closest to creeks. (2) The elevations of connected and unconnected bare patches tend to be lower relative to the tidal frame in sites with a smaller tidal range, although our analysis only included three sites. (3) Recovery of vegetation
in bare patches at the time scale of the last two decades was only observed in the site with high tidal range and high sediment input. Here vegetation recovery is hampered by low surface elevations relative to the local tidal frame, by farther distance from channels and by a high connectivity to the channel network. Below we will further substantiate these findings, and discuss interpretations and potential hypotheses that may explain mechanisms of formation and recovery of bare patches.



## 6.1 Topographic conditions determining the presence of bare patches

Our results suggest that bare patches exist under qualitatively similar topographic conditions across three different marsh systems. We found that marshes have a higher elevation than bare patches, in accordance with previous studies (DeLaune et al., 1994; Erwin et al., 2006; Wilson et al., 2014). In addition to existing insights, we found that unconnected bare patches are most frequently found at higher elevations and shorter distances from channels as compared to connected bare patches (Figs. 4 and 5). Additionally, we found a positive relationship between patch size and the width of the connecting channel

(Fig. 6). These different observations may be interpreted as follows. First, the positive relationship between bare patch size and connecting channel width may be due to the difference in tidal prism (i.e. the total water volume that floods into, and drains out of, the bare patches during a tidal cycle). A larger bare patch implies a larger tidal prism, which means that higher volumes of water are transported into and out of the bare patches. Assuming that most of the water is transported through the connecting channel, a larger tidal prism would be associated with larger channel-forming discharges and therefore wider

channels (e.g., Rinaldo et al., 1999b; Kirwan et al., 2008; D'Alpaos et al., 2010; Vandenbruwaene et al., 2013).

Secondly, our finding that unconnected bare patches occur most frequently on higher elevations than connected bare patches, may be interpreted as follows. We expect that connected bare patches experience higher incoming and outgoing flood and ebb flow velocities as they are directly connected to the channels. Furthermore, we found that connected bare patches are larger (Fig. 6), and hence we may expect more potential for erosion of surface sediments induced by waves (because of

larger wind fetch length). Wave erosion in interior marsh ponds has been found to be related to the size and wind fetch length of marsh ponds (Mariotti and Fagherazzi, 2013; Mariotti, 2016; Ortiz et al., 2017). Hence larger bare patches are likely to experience more wave-induced erosion and are found in this study to be connected through wider connecting channels, which may facilitate the tidal export of the eroded sediments from connected bare patches, and therefore may explain the lower surface elevation of connected bare patches. In contrast, we hypothesize that unconnected bare patches,

which are typically smaller (Fig. 6), may be expected to experience less wave erosion (smaller fetch length) and much weaker flow velocities (as flow is obstructed by surrounding vegetation). With respect to the latter effect, we notice that our classification of unconnected patches may also include patches with connecting channels smaller than 0.5 m but impossible to detect on the aerial images. Nevertheless, also in the case of such small connecting channels < 0.5 m wide, one can expect that drainage of the bare patches after overmarsh tides is much slower, with lower ebb flow velocities, as compared to bare

patches with wide connecting channels (up to several tens of meters wide, see Fig. 6), facilitating faster drainage, higher ebb flow velocities and potentially leading to larger tidal export of eroded sediments. Unconnected bare patches were also found to occur most frequently at shorter distances from channels as compared to connected bare patches, and this may facilitate higher sediment supply to unconnected bare patches closer to channels, as suspended sediment concentrations typically decrease with increasing distance from channels (Leonard, 1997; Christiansen et al., 2000; Temmerman et al., 2003b).

Therefore, higher sediment supply and lower magnitude of waves and tidal currents in smaller, unconnected bare patches at shorter distance from channels, may facilitate the settlement of suspended sediments and reduce erosion, and as such may



explain our finding of higher surface elevations of unconnected bare patches as compared to connected bare patches (Fig. 9). This finding is also in accordance with the model of Mariotti (2016) proposing that, in what is called the "pond collapse regime", the depth of connected marsh ponds would be larger than the depth of unconnected ponds.

Thirdly, our results indicate that connected bare patches are predominantly located farther away from channels than unconnected bare patches (Fig. 5). One potential explanation is that connected bare patches are generally larger than unconnected bare patches (Fig. 6), so that a larger fraction of the connected bare patches is located at a farther distance from channels. Presence of bare patches in relation to distance from channels has been previously studied on large regional scales ($10^2$–$10^4$ m) considering only large estuarine channels (Turner and Rao, 1990; Kearney and Rogers, 2010). On a smaller

scale (10–$10^2$ m), Redfield (1972) qualitatively reported that big bare patches are located relatively far from channels. Adamowicz and Roman (2005) observed that bare patches were located at around 11 m from the nearest channel in both ditched and unditched marshes in New England. Such a value is similar to that found for Saeftinghe and San Felice, but smaller than the value obtained for Blackwater. The elevation difference between connected and unconnected bare patches probably relates to their difference in distance to channels. Marshes typically have a micro-topography of higher levees

along channels and lower depressions farther away from channels as a consequence of progressive suspended sediment deposition during tidal flooding of marshes from channels (e.g., Reed, 1988; Covi and Kneib, 1995; Leonard, 1997; Esselink et al., 1998; Reed et al., 1999; Allen, 2000; Temmerman et al., 2004; D'Alpaos et al., 2007; Bartholdy, 2012). In accordance with this micro-topography, the lower-elevation connected bare patches are located farther away from channels than the higher-elevation unconnected bare patches (Fig. 9). In addition, the frequency distribution of distance to the closest channels

is observed to be exponential for the vegetated marsh surfaces in all three marsh sites, which is analogous to the results by Marani et al. (2003) and holds only for the vegetated marsh surfaces.

Finally, our results demonstrated that the size of bare patches is negatively related to the number of bare patches (Fig. 6). Such finding has also been observed in other marsh systems (Turner and Rao, 1990; Schepers et al., 2017). This may be indicative for initial formation of many small bare patches that grow and merge together through time, hence leading to a

decreasing number of larger patches. This process of merging of initially small bare patches into larger patches has been documented for the Blackwater study site from an analysis of time series of aerial pictures over the period 1938-2010 (Schepers et al., 2017).

In conclusion, we observed qualitatively similar topographic conditions for the presence of bare patches across the three study sites, albeit that the elevations of connected and unconnected bare patches tend to be lower relative to the tidal frame in

sites with a smaller tidal range. The latter agrees with earlier findings that micro-tidal marshes have in general a lower surface elevation than macro-tidal marshes (Kirwan et al., 2010; D'Alpaos et al., 2011). Our finding suggests that feedback mechanisms between vegetation and topography are important in regulating the position of the bare patches, and perhaps generalizable across systems. However, we emphasize that our analysis is based on only three study sites, and more research is needed to assess the degree to which this finding is universal.





### 6.2 Topographic conditions determining the revegetation of bare patches

The comparison between bare patches with two extreme revegetation rates (i.e., permanent bare patches over the studied 18-year period and rapidly revegetated bare patches within 4-6 years) for Saeftinghe suggests that fast revegetation preferentially occurs by expansion of the vegetated edge into small, higher elevation, unconnected bare patches, whereas the central areas of big, lower elevation, connected bare patches tend to remain unvegetated over the considered time period of 18 years. These results are consistent with previous studies. For example, only small bare patches were invaded by vegetation in ditched marshes in Louisiana, although large bare patches were permanent over a study period of 22 years (Turner and Rao, 1990). In several New England marshes, re-establishment of vegetation started within 1-2 years after unconnected bare patches merged with the channel network and became drained (Wilson et al., 2009, 2014). Additionally, some studies find that unconnected bare patches expand and merge quickly, while connected bare patches are relatively stable (Kearney et al., 1988). These disparate observations in different marsh sites may be due to different environmental conditions, such as differences in relative sea level rise and sediment availability (Mariotti, 2016). In a modelling study, Mariotti (2016) demonstrated that vegetation recovery in marsh ponds is favoured under conditions of slow relative sea level rise, large tidal range, and large inorganic sediment supply.

### 6.3 Vulnerability for bare patch formation and resilience for bare patch recovery

Although our study includes only three marsh sites, differences in tidal range, sediment availability, and rate of relative sea level rise (RSLR) between the sites allow us to explore their impact on marsh vulnerability and resilience in terms of bare patch formation and recovery. As explained in the description of the three study sites (see Sect. 2), the average tidal range and suspended sediment concentrations vary from highest in the Saeftinghe marsh (4.9 m and 30-60 mg/l respectively), intermediate in San Felice marsh (1 m and 10-20 mg/l), to lowest in the Blackwater marshes (0.5 m and 50 mg/l). Long-term RSLR rates in the San Felice and Blackwater marshes are within the same range of 3–4 mm/yr, while mean high water level rise in Saeftinghe is 5.7 mm/yr. In Saeftinghe, marsh elevations are mostly above MHWL, while they are mostly below MHWL in San Felice and Blackwater (Fig. 4). Probably this may explain why the proportion of bare surface area is larger in San Felice and Blackwater (34.33 % and 42.58 %, resp.) than in Saeftinghe (15.72 %). This may also suggest that marshes are more prone to presence of bare patches in sites with lower tidal range and sediment availability (Figs. 1-3), and is furthermore in accordance with the modelling study of Mariotti (2016), demonstrating that pond formation increases and pond recovery decreases with decreasing sediment availability, decreasing tidal range, and increasing rate of RSLR. Based on the comparison of our three study sites, we could not draw clear conclusions on the role of RSLR rate.

Marsh resilience inferred by revegetation of bare patches was only observed in Saeftinghe where the mean tidal range is 4.9 m, and bare patches have high elevations relative to the tidal frame (average RE = 1.002 for connected bare patches and average RE = 1.02 for unconnected bare patches; Fig. 4). Revegetation of bare patches has been observed in other systems with high tidal ranges (Millette et al., 2010; Wilson et al., 2014), which facilitates well-drained conditions during low tide





and enables vegetation regrowth. In contrast, in Blackwater where the mean tidal range is about 0.5 m, bare patches have a much lower elevation relative to the tidal frame, even below the MLWL (average RE = -0.462 for connected bare patches and average RE = 0.284 for unconnected bare patches; Fig. 4), which means that there is no drainage at low tide so that marsh vegetation cannot recover. Bare patches also tend to be permanent in other systems under low tidal ranges, such as in Louisiana and mid-Atlantic US salt marshes (Wilson et al., 2014; Ortiz et al., 2017). Clearly, the same elevation loss in a marsh with small tidal range will result in a higher increase in tidal inundation frequency and duration, and consequently in more stress on vegetation growth, as compared to a marsh with a large tidal range. Hence, if marsh vegetation and elevation loss occur, it would be easier to recover for marsh vegetation in a higher tidal range environment, such as that of Saeftinghe, as compared to situations with a lower tidal range, such as the Blackwater and San Felice marshes. This interpretation is in agreement with previous studies. Microtidal marshes were reported to be particularly vulnerable to bare patch formation and expansion (Kearney et al., 1988; Mariotti and Fagherazzi, 2013). Marshes with larger tidal ranges also have bare patches but they are generally more dynamically forming and recovering, while the whole marsh system is relatively stable (Redfield, 1972; Wilson et al., 2009). The model of Kirwan and Guntenspergen (2010) suggested that extensive bare patches occur, expand quickly and become permanent under small tidal ranges but not under large tidal ranges, because the elevation range suitable for vegetation growth is smaller in low tidal range environments. In general, marsh stability is positively related to tidal range (Kirwan et al., 2010; D'Alpaos, 2011), and numerical modelling indicates that high sediment concentrations are necessary for recovery of bare patches (Mariotti, 2016). However, Mariotti (2016) only considers recovery after connection to the tidal channel network, not the recovery of isolated bare patches. In our study, we observed that bare patches unconnected to the tidal channel network all recovered at the Saeftinghe site. Complete drainage of the Saeftinghe bare patches during ebb tides might explain this apparent discrepancy. We suggest that the close distance to channels (see Figs. 5 and 8b) (e.g., Ursino et al., 2004) and coarser sediment associated with channel levees (Allen, 2000) enable the unconnected bare patches to drain completely in Saeftinghe through subsurface drainage, and allow vegetation recovery.

### 6.4 Marshes and bare patches as two stable states in intertidal areas

High-elevation, vegetated marsh areas and low-elevation, unvegetated tidal flat areas have been previously identified as alternative stable states, primarily at the large scale of whole tidal basins (km²) (e.g., Fagherazzi et al., 2006; Marani et al., 2007, 2010; McGlathery et al., 2013; Wang and Temmerman, 2013; Moffett et al., 2015). Field evidence for alternative stable state behavior is however scarce for the smaller scale of vegetated and bare patches within marshes (10–100 m²).

Here we evaluate and discuss our results within the framework of alternative stable state theory, based on the conceptual model presented in Fig. 9. The state variables that are considered are (1) the vegetation biomass (high for the vegetated state versus zero for the bare state) and (2) the surface elevation. According to field studies and models of marsh evolution, both state variables interact through feedback loops: presence of vegetation will promote accretion of mineral and organic sediments, therefore resulting in higher surface elevation (Neumeier and Amos, 2006; Mudd et al., 2010; Vandenbruwaene et al., 2011; Yang et al., 2012; Hu et al., 2014), which further stimulates biomass production, up to a point where a high





equilibrium elevation is reached where biomass productivity reaches an optimum (Morris et al., 2002; D'Alpaos et al., 2007; Kirwan and Murray, 2007; Marani et al., 2007, 2010); while vice versa, absence of vegetation will facilitate erosion induced by tidal currents and waves, hence lowering surface elevation, which further prohibits vegetation growth, until a low bare equilibrium elevation is reached where erosion and deposition by waves and tidal currents are in dynamic equilibrium (Fagherazzi et al., 2006; Kirwan and Murray, 2007; Marani et al., 2007, 2010). Such internal feedback loops between

vegetation and elevation, i.e. the state variables, are indicated along the Y-axis in Fig. 9b, and can lead to two stable states, a high-elevation vegetated state (the green curve in Fig. 9b) or low-elevation bare state (the grey curve in Fig. 9b). The conditions under which these two states can develop, are indicated along the X-axis of Fig. 9b, such as the sediment supply and level of soil drainage/aeration. Previous modelling studies have shown that high sediment supply and low sea level rise rates (i.e. allowing long periods of intertidal soil drainage and aeriation) favors the evolution towards a high vegetated state,

while low sediment supply and high sea level rise rates (i.e. leading to shorter periods of intertidal soil drainage and aeration) promote formation of the low bare state (Marani et al., 2007, 2010; Mariotti, 2016). Within the context of our analysis, such conditions are spatially varying within marshes as a function of distance from tidal channels, as both sediment supply (e.g., Christiansen et al., 2000; Temmerman et al., 2003a) and soil drainage/aeration (e.g., Ursino et al., 2004) typically decrease with increasing distance from channels. Additionally, connectivity of bare patches to the channel network may further

influence sediment import into or export from bare patches, as discussed before.

Hence, we propose that at short distance from channels, conditions are favorable for a high, vegetated stable state, while at large distance from channels, conditions are favorable for a low, bare stable state; and that at intermediate distances from channels, both vegetated and bare states may co-exist next to each other as alternative stable states (Fig. 9a and b). Our results provide suggestive support for this hypothesis and conceptual framework, although we emphasize that our discussion here is very indicative and far from conclusive. Our results indicate that the occurrence of vegetated marshes peaks at the

highest elevations (Fig. 4) and shortest distances from channels (Fig. 5), which may be indicative for the high-elevation vegetated stable state (left part of Fig. 9b). Connected bare patches peak at the lowest elevation (Fig. 4) and farthest distances from channels (Fig. 5), and may be indicative for the low-elevation bare stable state (right part of Fig. 9b). At intermediate distances from channels, vegetated and bare marsh portions can exist next to each other (Fig. 5; and see e.g. Fig. 3). In the one field site with observed vegetation recovery (Saeftinghe), the bare areas that were never revegetated in the

studied 18-year period predominantly occurred in large connected bare patches at the lowest elevations and farthest distances from channels (Fig. 8). This may be indicative for the existence of bare stable states that are most frequently found at far distances from channels (right part of graph in Fig. 9b). In that same field site, fast revegetation (within 4-6 years) was predominantly observed in small unconnected bare patches at intermediate elevations and intermediate distances from

channels (Figs. 7 and 8). This may be indicative for disturbances of vegetation cover that quickly recover to the vegetated state, at intermediate distances (left part of graph in Fig. 9b). Comparison between the three study sites, suggests that conditions for the bare stable state are more likely in sites with lower tidal range (i.e. less soil drainage/aeration) and low





sediment supply, as revegetation was not observed in the two sites with lowest tidal range and sediment supply (Blackwater and San Felice), and only occurred in the site with highest tidal range and sediment supply (Saeftinghe) (Fig. 9b).

Finally, we note that the bio-geomorphic feedback mechanisms (i.e. vegetation-elevation feedbacks) leading to the elevated marsh state are similar in large-scale or small-scale studies, but there might be differences leading to the low, unvegetated state. In large lagoons or extensive tidal basins, the low elevation of the tidal flats is mainly maintained by feedbacks between wave erosion and water depth (Fagherazzi et al., 2006; Mariotti and Fagherazzi, 2010). Wave-induced erosion is, however, expected to be a negligible process in small bare patches such as those of Saeftinghe (max. 60 m wind fetch length)

that completely drain at ebb tide. Hence wave erosion is not likely to be a relevant mechanism controlling the depth of small bare patches or the evolution of small bare patches in Saeftinghe. Our results indicate that a connection with the tidal channel system is associated with a lower elevation of the bare patches, probably because of higher flow velocities that decrease sedimentation rates or cause erosion. Other work suggests that the substrate of bare ponds within marshes might be eroded and exported through the connecting channels (e.g., Day et al., 2011; Schepers et al., 2017). Nevertheless, further research

should test these hypotheses.

## 7 Conclusions

In this paper, we studied the topographical conditions for presence and revegetation of bare patches within three coastal marsh sites that are largely different in tidal range, sediment supply and plant species. The analyses of aerial photographs, LIDAR data and field topographic measurements showed that the topographic conditions (i.e. elevations, distances from

channels and connectivity to channels) for presence of bare patches were qualitatively consistent among the three marsh sites. We found that bare patches connected to channels occur most frequently at the lowest surface elevations and farthest away from creeks; unconnected bare patches most frequently occupy intermediate elevations and distances from creeks, and are smaller in size and larger in number; and vegetated marshes dominate at the highest surface elevations and closest to creeks. Further, we showed that the elevations of connected and unconnected bare patches tend to be lower relative to the tidal frame

with increasing tidal range, although our analysis only included three sites. Revegetation of bare patches was only observed in one site, which was the site with the highest tidal range and the largest sediment supply. For that site, we found that the chance of bare patch revegetation decreases with increasing width of channels that connect bare patches to the tidal channel network. The latter is associated with lower bare patch elevation, farther distance to channels and bigger bare patch size. Finally, in the context of sea level rise, our results suggest that the marsh site with the highest tidal range and highest

sediment input is less vulnerable to bare patch formation and more resilient in terms of revegetation of bare patches than the two other marsh sites with lower tidal range and lower sediment supply.



**Data availability**

The aerial images and DTM data for Saeftinghe can be downloaded from Rijkswaterstaat (https://geoservices.rijkswaterstaat.nl). The IKONOS data for San Felice can be accessed at Planetek Italia s.r.l. (https://www.planetek.it). The field elevation data for San Felice can be accessed at Venice Water Authority in Italy (http://provveditoratovenezia.mit.gov.it). The aerial images for Blackwater can be downloaded from earthexplorer.usgs.gov. The LIDAR data for Blackwater can be downloaded from https://inport.nmfs.noaa.gov/inport/item/49781.

**Supplement link**

The supplement related to this article is available online at:

**Author contribution**

ST and CW designed the study. CW prepared the manuscript with contributions from all co-authors.

**Competing interests**

The authors declare that they have no conflict of interest.

**Acknowledgements**

We dedicate this paper to the late Jose Busnelli who contributed to parts of the analyses presented in this paper. We thank Rijkswaterstaat (Dick De jong) in the Netherlands, Venice Water Authority in Italy, and United States Department of Agriculture and Maryland Department of Natural Resources in the USA for providing vegetation maps, aerial photographs, as well as elevation data of LIDAR surveys and field surveys. We also would like to thank Glenn Guntenspergen, Melissa Duvall, Patrick Brennand and Kyle Derby for the field measurements in the Blackwater Marshes.

**Financial support**

This work has been supported by the Project 41501116, 51761135022, and 41401413 supported by National Natural Science Foundation of China, the European Union Programme Erasmus Mundus External Cooperation Window (EMECW)–Lot 14–China and by the FWO research community "Functioning of River Ecosystems by Plant-Flow-Soil interactions", Research Foundation Flanders (FWO PhD grant L.S., 11S9614N), the Technology Foundation for Selected Overseas Chinese Scholar from Ministry of Personnel of China (2015), and the U.S. National Science Foundation (1237733, 1426981, 1654374, 1832221 and 1529245).



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



**Table 1. Overview of bare patch number (bare patches smaller than 1 m$^2$ were excluded from the analysis), Lidar pixels and GPS measurements of the three field sites.**

| Field site | Type | Number of bare patches | Number of Lidar pixels | Number of GPS measurements |
|---|---|---|---|---|
| Saeftinghe (NL) | Marsh | - | 67729 | - |
| | Unconnected bare patch | 97 | 1722 | - |
| | Connected bare patch | 58 | 12651 | - |
| San Felice (IT) | Marsh | - | 361261 | 340 |
| | Unconnected bare patch | 70 | 2556 | - |
| | Connected bare patch | 124 | 260140 | 95 |
| Blackwater (USA) | Marsh | - | 184871 | 93 |
| | Unconnected bare patch | 255 | - | 36 |
| | Connected bare patch | 227 | - | 31 |



**Figure 1: Study area in the Saeftinghe marsh.** (a) Location within Northwest Europe. (b) False-color aerial photograph of 2004 for the
study area. (c) Spatial distribution of vegetated marshes, unconnected bare patches and connected bare patches in 2004 with LIDAR
images as background.



Earth **Surface**
**Dynamics**
Discussions

**Figure 2: Study area in the San Felice marsh.** (a) Location within South Europe. (b) IKONOS image of 2006 for the study area. (c) Spatial distribution of vegetated marshes, unconnected bare patches and connected bare patches in 2002 with LIDAR images as background.






**Figure 3: Study area in the Blackwater Marshes.** (a) Location within the USA. (b) False-color aerial photograph of 2010 for the study area. (c) Spatial distribution of vegetated marshes, unconnected bare patches and connected bare patches in 2010 with LIDAR images as background.



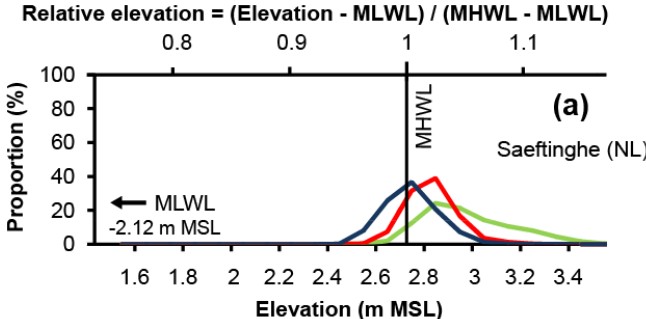

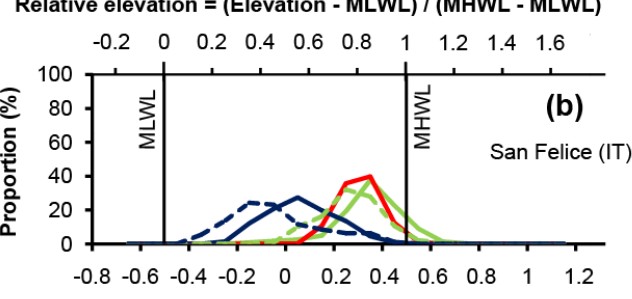

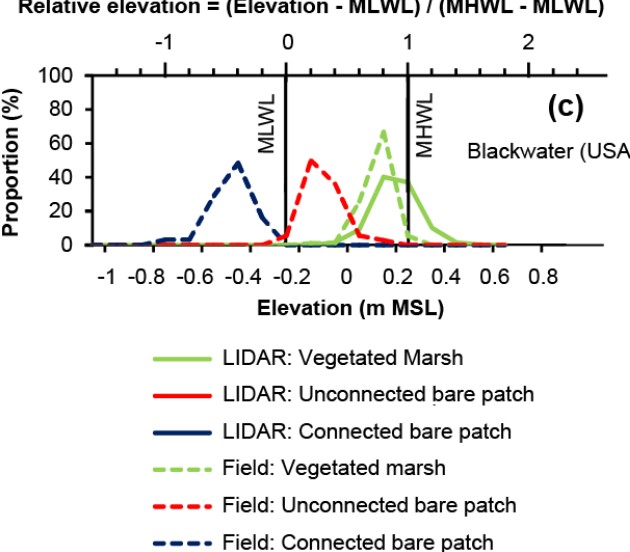


**Figure 4: Elevation distribution of vegetated marshes, unconnected bare patches and connected bare patches based on LIDAR surveys and field surveys for (a) Saeftinghe, (b) San Felice and (c) Blackwater.** The elevation on the bottom main X-axis is relative to the local mean sea level (i.e., m MSL). The relative elevation on the top secondary X-axis is rescaled as a proportion of the local tidal range (see Eq. (1)). The proportion on the Y-axis is calculated based on LIDAR or field measurements as the number of pixels or samples

in each elevation class (every 0.1 m) relative to the total number of pixels or samples for each feature. The exact numbers in each category are given in Table 1. The MLWL in Saeftinghe is 2.12 m lower than MSL, which is outside of the range of the main X-axis in Figure 4a. MLWL and MHWL definitions at Blackwater are approximate since water level changes are dominated by meteorological rather than astronomical influences.





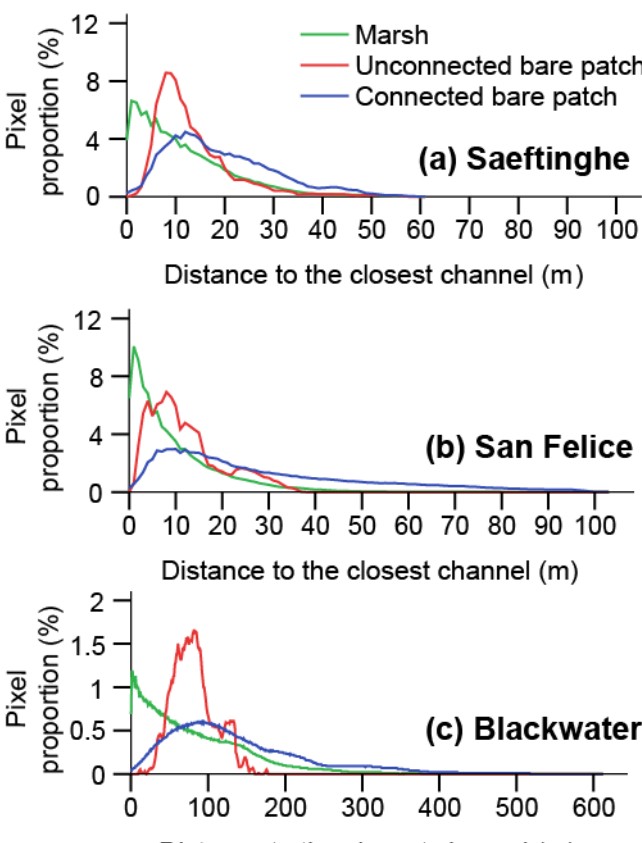

**Figure 5: Frequency distribution of distances to the closest channel in (a) Saeftinghe, (b) San Felice, and (c) Blackwater.** The proportion is calculated as the number of pixels in each distance class (every 1 m) relative to the total number of pixels for each feature, i.e., vegetated marshes, unconnected bare patches or connected bare patches.



Earth **Surface**
**Dynamics**
Discussions



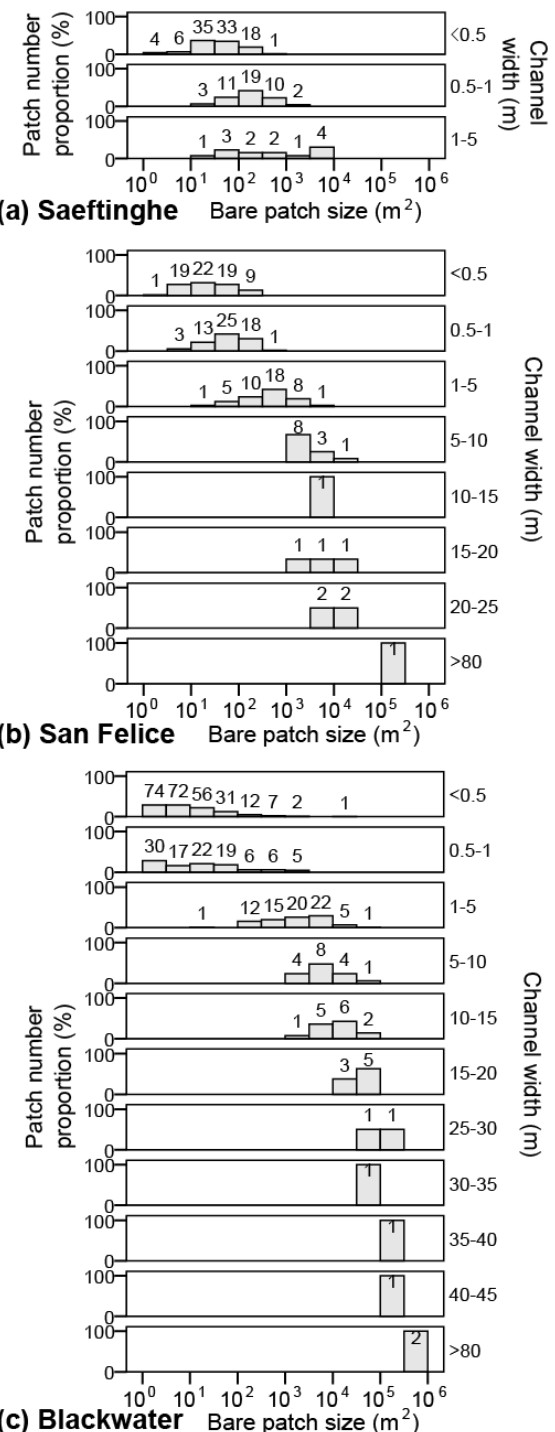

**Figure 6: Frequency distribution of bare patch sizes in relation to the connected channel width in (a) Saeftinghe, (b) San Felice,**
**and (c) Blackwater.** Note the X-axis is in logarithmic scale. The patch number proportion is calculated as the number of bare patches in
each class of bare patch size relative to the total number of bare patches for each category of channel width. The number of bare patches in
each size class is labeled at the top of the bars.



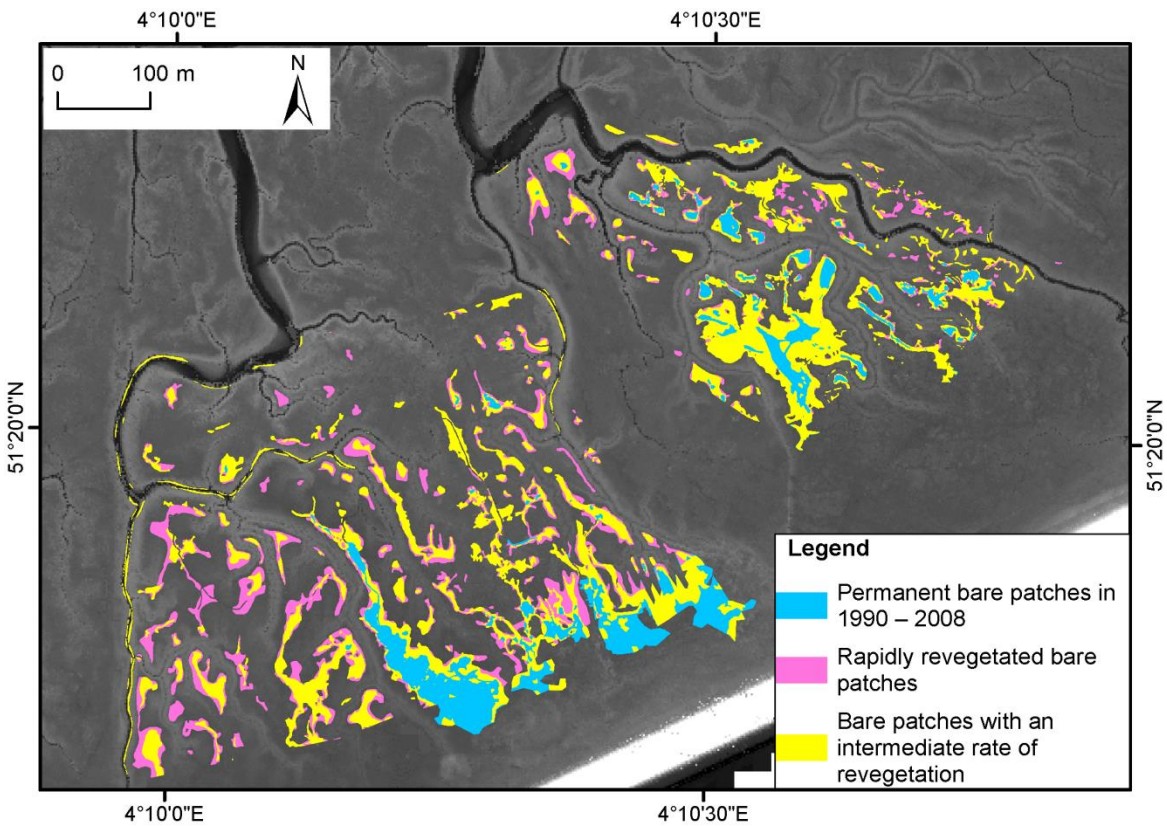

**Figure 7: Spatial distribution of bare patches with a different rate of revegetation in the period of 1990–2008 in Saeftinghe.**



Earth **Surface**
**Dynamics**
Discussions
EGU

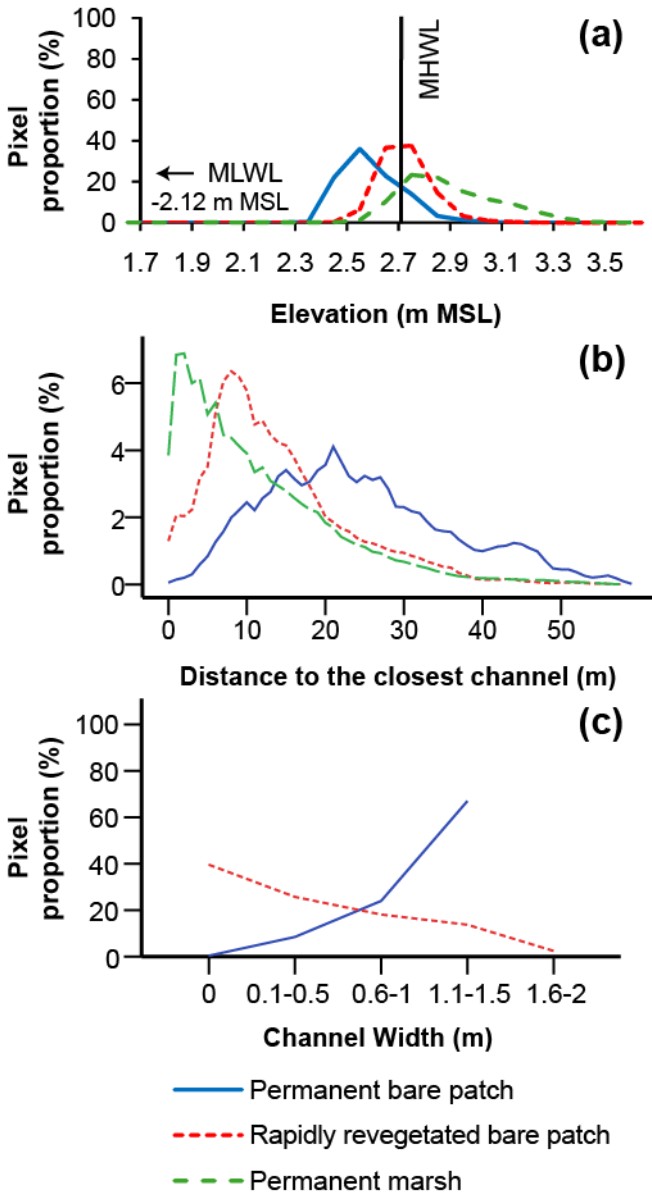

**Figure 8: Frequency distribution of (a) elevation, (b) distances to the closest channel, and (c) the connected channel width for permanent bare patches, rapidly revegetated bare patches and permanent marsh areas.** The elevation is relative to mean sea level and binned into 0.1 m intervals. The proportion in panel (a) is calculated as the number of pixels in each elevation class (every 0.1 m) relative to the total number of pixels for each feature. The proportion in panel (b) is calculated as the number of pixels in each distance class (every 1.0 m) relative to the total number of pixels for each feature. The proportion in panel (c) is calculated as the number of pixels in each class of channel width relative to the total number of pixels for each feature.



Earth **Surface**
**Dynamics**
Discussions

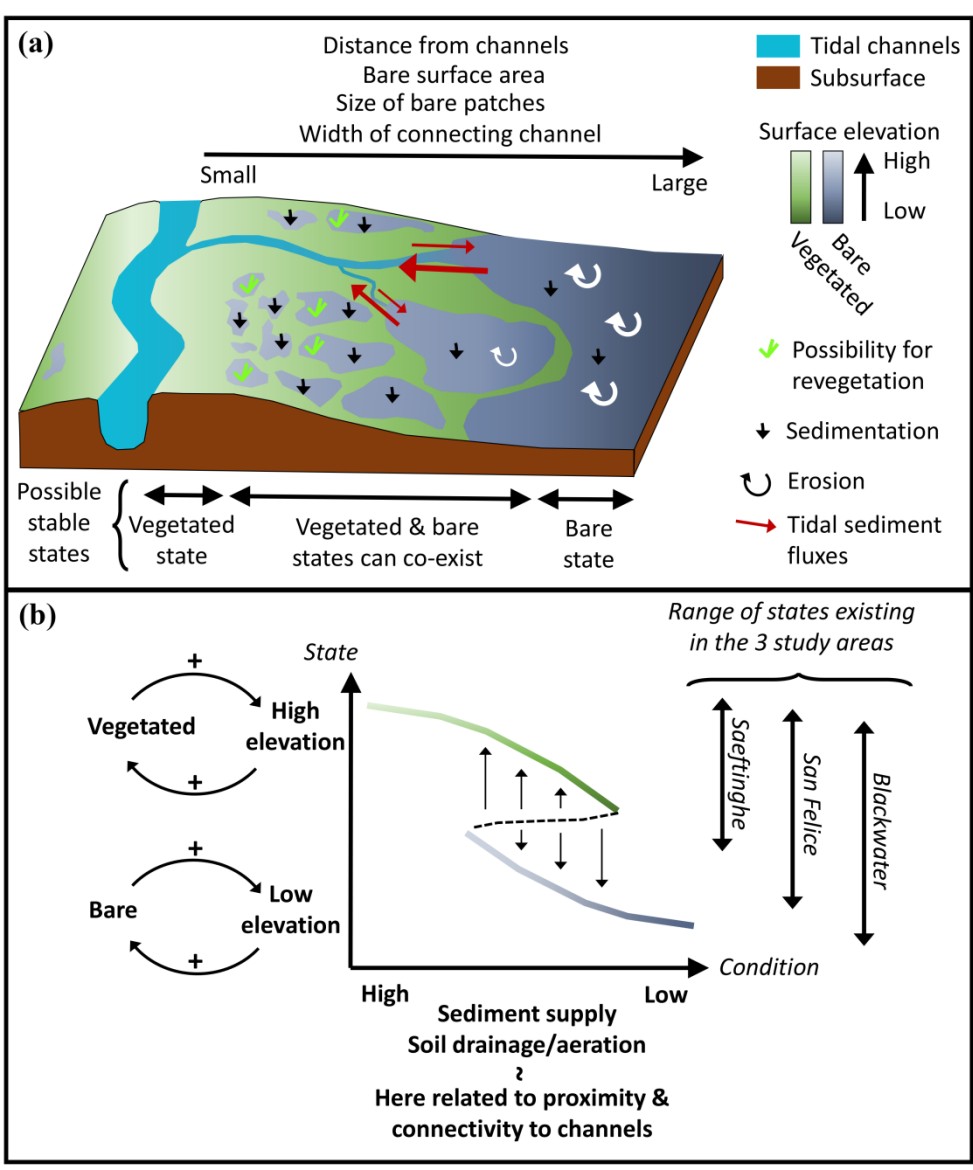

**Figure 9: Conceptual model summarizing results and interpretations.** In (a) the size of the arrows is indicative for hypothetical
magnitudes of sediment accretion (black arrows), erosion (white arrows) and tidal sediment fluxes (red arrows). The hypotheses are that
the larger bare patches connected to the channels experience stronger incoming and outgoing tidal currents, and more waves during
flooding (longer wind fetch length), favoring erosion and tidal export of sediments via the channels; while the smaller unconnected bare
patches experience weak tidal currents (because of obstruction by surrounding vegetation), few or no waves (small wind fetch length),
limiting erosion and allowing accretion. In (b) we interpret the results within the framework of the alternative stable state theory, using a
hypothetical/conceptual plot of states (Y-axis) versus conditions (X-axis). State variables are vegetation biomass (high for the vegetated
state = the green curve, versus zero for the bare state = the grey curve) and surface elevation (different shades of green and grey).
Conditional variables are sediment supply and soil drainage/aeration, which are in our framework spatially varying with distance and
connectivity to tidal channels.