# Peer review of "Different coastal marsh sites reflect similar topographic conditions under which bare patches and vegetation recovery occur"

_Earth Surface Dynamics, 2020_

## Referee Comment (RC1) · Anonymous Referee #1 · 23 Jul 2020

This paper is generally well written, studied the topographical conditions determining the presence and revegetation of bare patches in three marsh sites with contrasting tidal range, sediment supply and plant species, respectively distributed in three countries. The introduction is detailed, the methodology is well described, the results are clearly described, and the discussion is well-founded and consistent with current knowledge of the subject. Even so, I suggest some minor changes that I detail below, which I hope would be useful to improve this paper. 1. The information about some materials and methods should be more detailed and clear in the abstract. 2. The Study area, Materials and data preprocessing, and Data analysis are too long, it's better to make them a bit more concise. 3. It's better to detail some of the implications of the results

and some useful advice to the policy maker in the discussion. 4. I suggest to move some figures to the supporting information, as there are too many figures in the main text now.

**ESurfD**

---

## Referee Comment (RC2) · Maarten Kleinhans (Referee) · 26 Jul 2020

**1   Main comments**

This manuscript presents a data analysis of bare patches in saltmarsh, in particular of the causal variables deemed to govern their formation and possible revegetation. Three different systems are analysed with different tidal ranges and sediment availability. Two main conclusions seem not sufficiently well supported.

The first is that sediment availability and tidal range determine the potential for revegetation, but three study areas are insufficient to isolate one of these two variables, let
alone assess their effect in combination. Two of the areas have low sediment availability and the one area with more sediment also has the highest tidal range.

The second conclusion is that the appearance and possible disappearance of unvegetated patches in saltmarsh systems are acting as a bistable state system. While this concept is currently in fashion, the work done here is of interest in its own right and there appears no other support for the idea in this paper than the frequent use of it in other saltmarsh papers.

Furthermore there are some unanswered questions, such as whether inundation duration would not be a more appropriate biophysical boundary condition than the elevation in the tidal frame. A number of the variables that the study refers to, such as sediment availability, are not measured.

Finally work needs to be done on the figures for a clearer presentation of the data and its context. These issues together, further detailed below, suggest that a moderate revision is needed.

**2 Detailed comments**

**2.1 Preamble and conclusions**

The title does not reflect the contents and is ambivalent (do the similar topographic conditions refer to different coastal marsh sites or to bare patches and vegetation recovery?)

The abstract requires some clarification: the sentence "Our results demonstrate that ... distance from the main channels." Do the authors simply mean with 'across' that that all the sites show the same pattern? What kinds of channels are the patches connected to, since these are furthest away from the main channels (whatever they are)?

The conclusion of the abstract that bare patches may form rapidly and become vegetated rapidly in the unstable zone at intermediate channel distances is based on only one of the sites, which begs the question whether the proposed existence of two stable states can be supported by the data, and how those bare patches at the other sites came about. Were they always unvegetated? Did they die off when the inundation duration increased, as the saltmarsh developed and reduced the outflow at these locations?

Line 387 provides an interpretation of why the bare patches sit on higher elevations. This is based on expectations (meaning inferences without evidence), rather than measurement, and not even basic calculations (or readings from the classic wind waveheight plots on the basis windspeed, fetch (here patch size) and depth) are provided. Possibly the ideas here are biased by the reviewed literature as well and other alternative hypothesis could explain the observations. In Brückner et al. (2019, https://doi.org/10.1029/2019JF005092, also situated in the Western Scheldt) the modelling shows that expanding saltmarsh may, counterintuitively, lead to increased inundation duration within the marsh, which then leads to die-off. Indeed, the elevation within the tidal frame (as used here) may not be the appropriate measure. I wonder what the inundation duration, or perhaps the hydrodynamic energy, is at the elevations of the connected and the disconnected bare patches, and whether too long inundation has to do with the die-off (assuming these patches were vegetated before), as suggested in Brückner et al. This also fits with the observation that sediment supply is needed to lift up the area and reduce inundation for revegetation.

**2.2 Comments on results**

Figures 1 to 3 show insufficient context. One key variable for the authors is connectivity of the patches in terms of distance to channels, so the bigger context of the study areas must be show to see the bigger and smaller channels. This would be more interest-
ing information than the photographs in the panels (which have different meanings for colour anyway). An image or lidar map showing the surrounding landscape including the channels would be more useful here, and the original images can go to the online supplement. For Saeftinghe I checked and the study location in Figure 7 is quite close to the embanked boundary of the system (so the white band on the bottom left is in fact an embankment). In fact the right zone is quite close to an old embankment within the area and one wonders whether that leads to enhanced ponding and a modified channel pattern like one can see further east along the dike.

Why are there bare patches not considered in Fig. 3?

Figure 4 has a lot of redundant header and axis text information and the real information is hidden on a few square centimeters. Likewise for Figure 5, where removal of the horizontal axis texts for panels a and b makes it possible to have higher plots on the same space, so that the data are more clearly shown and comparable. This is necessary, because what happens in the tails of these skewed distribution is interesting: the connected bare patches plot above the other distributions.

Figure 6 contains novel information and shows interesting trends. However, the relative vertical axis per channel width class leads to a biasing emphasis on a very small number of cases for the largest channel widths. Perhaps another presentation would solve that problem: a matrix (pcolor in matlab) with log(patch size) on the horizontal axis, log(channel width) on the vertical, and log(number or fraction of total) as the colour scale. The channel width classes are not consistent with the possibly logarithmic distribution of the number of patches against channel width and I suggest to simply use classes of a 2-base log or something here, which would also improve the horizontal axis in Figure 8c from non-equidistant class to a true width scale.

Figure 8 needs to mention in the caption that this concerns the Saeftinghe site only. Is distance to the closest channel calculated from a map of channels or from the DEM? How is the information in panel c obtained; is that the same as in Figure 6a but then

split up for the permanent and temporary bare patches? Why is there no data for the other areas? I suppose there are older images so this is open for analysis. As it stands now, there is very little data and support for the conclusions about stability and revegetation, especially since this plot is only for the system where the authors claim that revegetation is most likely. How do they know?

Figure 9a has four variables mentioned on the top arrow to the right, but is width of the connecting channel really increasing to the right, away from the widest main channel and into the bifurcating network? That is only possible if the reduction in depth goes much more rapid. Is erosion the right term here? How is it possible that sediment disappears in such a strongly converging flow (meaning very low velocity in the patches landward of the first bit of well-defined channel)? Are waves important here, as high up on the marsh in a very shallow, vegetated and micro-fetch area? Waves are known to be important in this sort of system, but that is on saltmarsh edges where there is fetch and depth to generate waves. It is not simply saltmarsh collapse and disappearance of organic material that causes the bare patches?

In Figure 9b, the horizontal axis provides two complex variables: sediment supply and soil drainage, but how do you know that it concerns these two and not the many others mentioned in lines 77-80? These are entirely inferred here but not measured. Any concentration from literature such as in line 125 is meaningless because of the very large spatial variation and the sediment settling in the marsh so far from the channels. So the position of the blue and green curves in the graphic is really unknown and we cannot know whether there are really two disconnected lines or simply a single continuum. And that means that the connection to the bistable state diagram is entirely speculative. I know it is attractive to try and see the landscape through the filter of the concept from complexity theory (citations here go back to Scheffer but the idea is already reviewed in Thorn and Welford 1994 https://www.jstor.org/stable/2564149), but this connection needs to be supported by the data. At present, it is not, and removal of this panel and section 6.4 of the discussion would in my opinion increase the quality of

the paper.

The lidar images in the supplement are barely useful as presented here. The gray scale and small image size, and the lack of colour scale bar makes it very hard to see anything at all here.

**2.3 Suggestions for the text**

The present objective (line 108) is now to determine the topographic conditions determining the presence of bare patches, but the idea also seems to determine whether they can revegetate, so I suggest 'presence *and dynamics*'.

The authors define two kinds of bare patches, but surely this is a continuum and there is a certain image resolution. They need to indicate what size of connecting channel is the cutoff for an isolated or connected patch earlier than in line 397 in the discussion.

The size of bare patches is important for the discussion (line 427) but size is not plotted in Fig. 6, only number of pixels and that could also indicate many small patches. A plot of patch size, and possibly analyses with patch size as a variable, are needed to make this argument.

---

## Author Comment (AC1) · 29 Oct 2020

Dear Editor and Reviewers,

We would like to thank you for your handling and response to our submission. Please find enclosed the revised manuscript for Earth Surface Dynamics, entitled "Different coastal marsh sites reflect similar topographic conditions for bare patches and vegetation recovery" [Paper #esurf-2020-56], and detailed list of our responses to the comments of the two reviewers. We highly appreciated the comments made by the reviewers, as they enabled us to greatly improve the manuscript.

[Figure]

In the supplement, we give a step-by-step response to the comments. The original comments of the editor and reviewers are copied and shown in black. Our step-by-step replies are inserted and shown in blue. The line numbers that are mentioned refer to the line numbers in the revised manuscript with tracked changes.

Thank you very much for your continued consideration of this manuscript.

Looking forward to your reply.

Yours sincerely,

On behalf of all co-authors,

Chen Wang

Please also note the supplement to this comment:
https://esurf.copernicus.org/preprints/esurf-2020-56/esurf-2020-56-AC1-supplement.zip

---

## Author Response (AR1)

Editor

Earth Surface Dynamics

September 20, 2020

Dear Editor,

We would like to thank you for your handling and response to our submission. Please find enclosed the revised manuscript for *Earth Surface Dynamics*, entitled "*Different coastal marsh sites reflect similar topographic conditions for bare patches and vegetation recovery*" [Paper #esurf-2020-56], and detailed list of our responses to the comments of the two reviewers. We highly appreciated the comments made by the reviewers, as they enabled us to greatly improve the manuscript.

Below we give a step-by-step response to the comments. The original comments of the editor and reviewers are copied below and shown in black. Our step-by-step replies are inserted and shown in blue. The line numbers that are mentioned refer to the line numbers in the revised manuscript with tracked changes.

Thank you very much for your continued consideration of this manuscript.

Looking forward to your reply.

Yours sincerely,

On behalf of all co-authors,

Chen Wang

"**Different coastal marsh sites reflect similar topographic conditions for bare patches and vegetation recovery**" **[Paper #esurf-2020-56]**

Chen Wang, Lennert Schepers, Matthew L. Kirwan, Enrica Belluco, Andrea D'Alpaos, Qiao Wang, Shoujing Yin, and Stijn Temmerman

**List of response to the comments**

**1. Response to the comments from Reviewer #1**
This paper is generally well written, studied the topographical conditions determining the presence and revegetation of bare patches in three marsh sites with contrasting tidal range, sediment supply and plant species, respectively distributed in three countries. The introduction is detailed, the methodology is well described, the results are clearly described, and the discussion is well-founded and consistent with current knowledge of the subject. Even so, I suggest some minor changes that I detail below, which I hope would be useful to improve this paper.

**Response: We thank the reviewer for this overall positive evaluation.**

(1) The information about some materials and methods should be more detailed and clear in the abstract.

**Response: This is addressed on lines 19-21 of the revised manuscript (the version with tracked changes), which are modified to emphasize information on the materials and methods:**

**"Based on GIS analyses of aerial photos and LIDAR imagery of high resolution (≤ 2 x 2 m pixels), we analyzed the topographic conditions under which bare patches occur, including their surface elevation, size, distance to and connectivity to channels."**

(2) The Study area, Materials and data preprocessing, and Data analysis are too long, it's better to make them a bit more concise.

**Response: We shortened parts in the Sections of Study area, Materials and data preprocessing, and Data analysis. Especially the Study area was considerable shortened. We refer to the revised manuscript (the version with tracked changes) to see the text revisions.**

(3) It's better to detail some of the implications of the results and some useful advice to the policy maker in the discussion.

**Response: We added a paragraph in the discussion at the end of section 6.3 (line 523-527):**

**"Finally, our results may be indicative to decision makers on salt marsh management, as the formation of bare patches may be indicative for marsh degradation towards an unvegetated state that may be difficult to recover. Our study indicates that early signatures for marsh degradation must be particularly monitored in marsh portions, farthest away from main channels and with lowest surface elevations. Monitoring of early signatures is especially advised in systems with very low tidal range and suspended sediment availability."**

(4) I suggest to move some figures to the supporting information, as there are too many figures in the main text now.

**Response: We moved figure 7 to the Supplement (Figure S4). Panels (b) of Fig. 1-3 are also moved to the Supplement. Fig. 9 (b) is removed, as suggested by the next reviewer. We also re-arranged Fig. 1-3 to save space.**

**2. Response to the comments from Reviewer #2**
This manuscript presents a data analysis of bare patches in saltmarsh, in particular of the causal variables deemed to govern their formation and possible revegetation. Three different systems are analysed with different tidal ranges and sediment availability. Two main conclusions seem not sufficiently well supported.

The first is that sediment availability and tidal range determine the potential for revegetation, but three study areas are insufficient to isolate one of these two variables, let alone assess their effect in combination. Two of the areas have low sediment availability and the one area

with more sediment also has the highest tidal range.

**Response: Indeed we need to emphasize that we only investigated 3 sites, which it is not enough to fully assess the impact of site-differences, such as in tidal range and sediment supply, on occurrence and revegetation of bare patches. We only notice that revegetation only occurred at the site with largest tidal range and sediment supply, while it was not observed at the two other sites with smaller tidal range and sediment supply. But a much higher number of different sites should be investigated, to further assess the effects of tidal range and sediment supply on the occurrence and revegetation of bare patches.**

**We highlighted this in the discussion, by making the following modifications in the Discussion (section 6.3), lines 477-483:**

**"Previous modelling has suggested that pond formation increases and pond recovery decreases in marsh sites that are subject to a lower suspended sediment availability, smaller tidal range, and lower rate of relative sea level rise (RSLR) (Mariotti, 2016). First of all, we want to emphasize that we only investigated three sites, which it is not enough to fully assess the impact of site-differences, such as in tidal range, sediment supply, and rate of RSLR, on occurrence and revegetation of bare patches. Yet we notice that revegetation only occurred at the site (Saeftinghe) with largest tidal range and sediment supply, while it was not observed at the two other sites (San Felice and Blackwater) with smaller tidal range and sediment supply."**

**It was also highlighted in the conclusion, lines 600-603:**

**"However, we emphasize that our study only included three sites, and that further research comparing much more sites is needed, to further advance our understanding of why certain marsh sites are more vulnerable to others to formation and persistence of bare patches. Such knowledge will be important to inform decision makers on site-specific priorities for marsh conservation."**

The second conclusion is that the appearance and possible disappearance of unvegetated patches in saltmarsh systems are acting as a bistable state system. While this concept is currently in fashion, the work done here is of interest in its own right and there appears no other support for the idea in this paper than the frequent use of it in other saltmarsh papers.

**Response: The reviewer also comes back on this general comment in his more detailed comments below. We agree that our interpretation of results, in context of the alternative stable state theory, was inspired by previous literature on salt marshes, and interpretation of vegetated marshes and unvegetated tidal flats as alternative ecosystem states. We agree that, in this respect, the interpretation of our results is hypothetical. Therefore we followed the advice of the reviewer below, to remove section 6.4 from the discussion, and to remove panel (b) from Figure 9.**

**Accordingly, we also removed the last two sentences of the original abstract, where we summarized our interpretations in the context of the alternative stable state theory.**

Furthermore there are some unanswered questions, such as whether inundation duration would not be a more appropriate biophysical boundary condition than the elevation in the tidal frame. A number of the variables that the study refers to, such as sediment availability, are not measured.

**Response: These comments are further detailed below by the reviewer, and we will reply to them below.**

Finally work needs to be done on the figures for a clearer presentation of the data and its context. These issues together, further detailed below, suggest that a moderate revision is needed.

**Response: These comments are further detailed below by the reviewer, and we will reply to them below.**

**2 Detailed comments**

2.1 Preamble and conclusions

The title does not reflect the contents and is ambivalent (do the similar topographic conditions refer to different coastal marsh sites or to bare patches and vegetation recovery?)

**Response: We changed the title to make it more explicit, as follows:**

**"Different coastal marsh sites reflect similar topographic conditions under which bare patches and vegetation recovery occurs"**

The abstract requires some clarification: the sentence "Our results demonstrate that ... distance from the main channels." Do the authors simply mean with 'across' that that all the sites show the same pattern? What kinds of channels are the patches connected to, since these are furthest away from the main channels (whatever they are)?

**Response: We changed this sentence to make it clear (line 22-25):**

**"Our results demonstrate that, for the different marsh sites, bare patches can be connected or unconnected to the channel network, and that the width of the connecting channels increases with the size of the bare patches, in each of the three marsh sites. Further, pixels located in bare patches connected to channels occur most frequently at the lowest elevations and farthest distance from the channels."**

**Hence we added here that the analysis is done for individual pixels (which are ≤ 2 x 2 m, as explained in the previous sentence in the manuscript), and that the frequency distribution of these pixels, located in bare taches connected to channels, peaks at the lowest elevations and farthest distance from channels. This is because connected bare patches are also the largest patches (this was added in the first sentence above), containing much pixels far away from channels. We don't make a distinction between 'main' channels and 'other' channels, or whatsoever, but we analyzed the frequency distribution of distances to all channels. Hence we removed the word 'main' channels.**

The conclusion of the abstract that bare patches may form rapidly and become vegetated rapidly in the unstable zone at intermediate channel distances is based on only one of the sites, which begs the question whether the proposed existence of two stable states can be supported by the data, and how those bare patches at the other sites came about. Were they always unvegetated? Did they die off when the inundation duration increased, as the saltmarsh developed and reduced the outflow at these locations?

**Response: Also in response to a previous remark above, we significantly reduced our interpretation of the results in terms of the theory on alternative ecosystem states. Because indeed, in this respect, our interpretation is rather hypothetical. Accordingly, in the abstract, we also removed this reference to alternative stable states.**

Line 387 provides an interpretation of why the bare patches sit on higher elevations. This is based on expectations (meaning inferences without evidence), rather than measurement, and not even basic calculations (or readings from the classic wind waveheight plots on the basis windspeed, fetch (here patch size) and depth) are provided. Possibly the ideas here are biased by the reviewed literature as well and other alternative hypothesis could explain the observations. In Brückner et al. (2019, https://doi.org/10.1029/2019JF005092, also situated in the Western Scheldt) the modelling shows that expanding saltmarsh may, counterintuitively, lead to increased inundation duration within the marsh, which then leads to die-off. Indeed, the elevation within the tidal frame (as used here) may not be the appropriate measure. I wonder what the inundation duration, or perhaps the hydrodynamic energy, is at the elevations of the connected and the disconnected bare patches, and whether too long inundation has to do with the die-off (assuming these patches were vegetated before), as suggested in Brückner et al. This also fits with the observation that sediment supply is needed to lift up the area and reduce inundation for revegetation.

**Response: Indeed we discuss several possible hypotheses quite extensively in section 6.1., by referring to existing insights from the literature. We thank the reviewer for focusing our attention to the paper by Brückner et al., and added it in the hypothetical discussion in line 441-446:**

**"This micro-topography of levees close to channels and depressions further away from channels, is often associated with an increasing inundation duration after high tides, and decreasing soil drainage/aeration during low tides, with increasing distance from channels (e.g., Ursino et al., 2004). Also, a modelling study suggested that marsh vegetation expansion can lead to increased inundation time, and as such can feedback on increased stress and**

**chance for vegetation die-back (Br ückner et al. 2019). This may be all mechanisms that may contribute to increased chance for occurrence of bare patches within marshes at farther distances from channels."**

2.2 Comments on results

Figures 1 to 3 show insufficient context. One key variable for the authors is connectivity of the patches in terms of distance to channels, so the bigger context of the study areas must be show to see the bigger and smaller channels. This would be more interest- ing information than the photographs in the panels (which have different meanings for colour anyway). An image or lidar map showing the surrounding landscape including the channels would be more useful here, and the original images can go to the online supplement. For Saeftinghe I checked and the study location in Figure 7 is quite close to the embanked boundary of the system (so the white band on the bottom left is in fact an embankment). In fact the right zone is quite close to an old embankment within the area and one wonders whether that leads to enhanced ponding and a modified channel pattern like one can see further east along the dike.

**Response: We moved the aerial images (panels (b) in Figs. 1 to 3) to the online supplementary materials. Also, we added images of the larger surrounding landscape for the three study sites. We added these images also in the online supplementary materials, instead of in the main manuscript, as it contains already quite a lot of figures, and the other reviewer suggested reducing the number of figures in the main manuscript.**

Why are there bare patches not considered in Fig. 3?

**Response: This is explained in the methods section 3.4, line 237-241:**

**"In the Blackwater study site, we selected a study area away from the influence of roads and uplands (Fig. 3). The small study area (marked with shading in Fig. 3) was chosen for the field survey. A larger study area (the entire colored region in Fig. 3) was later considered in order to increase the number of bare patches connected to channels wider than 1 m. Bare patches that are connected with narrow channels (< 1 m) and that are located outside of the small study area (blue polygons in Fig. 3) were not considered in the analysis."**

Figure 4 has a lot of redundant header and axis text information and the real information is hidden on a few square centimeters. Likewise for Figure 5, where removal of the horizontal axis texts for panels a and b makes it possible to have higher plots on the same space, so that the data are more clearly shown and comparable. This is necessary, because what happens in the tails of these skewed distribution is interesting: the connected bare patches plot above the other distributions.

**Response: In Figure 4, we removed the label "relative elevation" at the upper X-axis for panel (b) and (c). Further, we removed the label "Elevation (m MSL)" at the bottom X-axis for panels (a) and (b). As a result we could make the figures a bit larger in the vertical direction.**

**In Figure 5, we followed the same advice, by removing the label "distance to closest channel (m)" from the horizontal axis, and making the figures a bit larger.**

Figure 6 contains novel information and shows interesting trends. However, the relative vertical axis per channel width class leads to a biasing emphasis on a very small number of cases for the largest channel widths. Perhaps another presentation would solve that problem: a matrix (pcolor in matlab) with log(patch size) on the horizontal axis, log(channel width) on the vertical, and log(number or fraction of total) as the colour scale. The channel width classes are not consistent with the possibly logarithmic distribution of the number of patches against channel width and I suggest to simply use classes of a 2-base log or something here, which would also improve the horizontal axis in Figure 8c from non-equidistant class to a true width scale.

**Response: We tried alternative representations of the results, in line with the suggestions made here, but it did not improve the representation. Actually, the pattern of increasing connecting channel width with increasing patch size, becomes less clear. Therefore, we chose to keep the same figure format.**

Figure 8 needs to mention in the caption that this concerns the Saeftinghe site only.

**Response: Indeed this was done.**

Is distance to the closest channel calculated from a map of channels or from the DEM?

**Response: Channels are mapped based on aerial images, which is explained in section 3.1**

How is the information in panel c obtained; is that the same as in Figure 6a but then split up for the permanent and temporary bare patches?

**Response: Indeed.**

Why is there no data for the other areas?

**Response: For the considered time periods (see sections 3.2 to 3.3) we did not observe revegation of bare patches in the other two areas (San Felice and Blackwater).**

I suppose there are older images so this is open for analysis. As it stands now, there is very little data and support for the conclusions about stability and revegetation, especially since this plot is only for the system where the authors claim that revegetation is most likely. How do they know?

**Response: Indeed our interpretations in the discussion, in the context of alternative stable state theory, were strongly hypothetical. Also in response to a previous similar remark by the reviewer, we left out this part of the discussion.**

Figure 9a has four variables mentioned on the top arrow to the right, but is width of the connecting channel really increasing to the right, away from the widest main channel and into the bifurcating network? That is only possible if the reduction in depth goes much more rapid.

**Response: Indeed, this may have been confusing. Therefore, the formulation was changed as**

**"connectivity of bare patches to channels". This is indeed small close to the main channel in the drawing, and increases with increasing distance from the main channel.**

Is erosion the right term here?

**Response: we rephrased it as "resuspension", as contrasting with the "sedimentation".**

How is it possible that sediment disappears in such a strongly converging flow (meaning very low velocity in the patches landward of the first bit of well-defined channel)? Are waves important here, as high up on the marsh in a very shallow, vegetated and micro-fetch area? Waves are known to be important in this sort of system, but that is on saltmarsh edges where there is fetch and depth to generate waves. It is not simply saltmarsh collapse and disappearance of organic material that causes the bare patches?

**Response: previous studies have proposed that waves play a significant role in resuspension (or erosion) of pond bottom material in marsh ponds, and in lateral expansion of ponds by wave-induced erosion of the pond-marsh boundaries (Mariotti, 2016; Ortiz et al. 2017). But these studies identified a critical interior marsh pond size of 200–1000 m for wave-induced erosion of vegetated pond edges (i.e. effect of fetch length). However, many of the bare patches considered in our study, are smaller, and hence wave-induced erosion is expected to be very small. Tidal currents are then the expected dominant control over hydrodynamics, eventually responsible for resuspension and export of sediments from bare patches.**

In Figure 9b, the horizontal axis provides two complex variables: sediment supply and soil drainage, but how do you know that it concerns these two and not the many others mentioned in lines 77-80? These are entirely inferred here but not measured. Any concentration from literature such as in line 125 is meaningless because of the very large spatial variation and the sediment settling in the marsh so far from the channels. So the position of the blue and green curves in the graphic is really unknown and we cannot know whether there are really two disconnected lines or simply a single continuum. And that means that the connection to the bistable state diagram is entirely speculative. I know it is attractive to try and see the landscape through the filter of the concept from complexity theory (citations here go back to Scheffer but the idea is already reviewed in Thorn and Welford 1994 https://www.jstor.org/stable/2564149), but this connection needs to be supported by the data. At present, it is not, and removal of this panel and section 6.4 of the discussion would in my opinion increase the quality of the paper.

**Response: We followed this advice, and removed panel (b) of Figure 9, and removed section 6.4 from the paper.**

The lidar images in the supplement are barely useful as presented here. The gray scale and small image size, and the lack of colour scale bar makes it very hard to see anything at all here.

**Response: We replace them by aerial images of the larger surrounding landscape.**

2.3 Suggestions for the text

The present objective (line 108) is now to determine the topographic conditions determining the presence of bare patches, but the idea also seems to determine whether they can revegetate, so I suggest 'presence and dynamics'.

**Response: we followed this suggestion.**

The authors define two kinds of bare patches, but surely this is a continuum and there is a certain image resolution. They need to indicate what size of connecting channel is the cutoff for an isolated or connected patch earlier than in line 397 in the discussion.

**Response: Indeed, we also mentioned this threshold channel width in the methods section 3.1 in line 186-189.**

The size of bare patches is important for the discussion (line 427) but size is not plotted in Fig. 6, only number of pixels and that could also indicate many small patches. A plot of patch size, and possibly analyses with patch size as a variable, are needed to make this argument.

**Response: Patch size is indeed plotted on the X-axis of Figure 6.**

[revised manuscript text omitted]

---

## Author Response (AR2)

Editor

Earth Surface Dynamics

December 22, 2020

Dear Editor,

We would like to thank you for your handling and response to our submission. Please find enclosed the revised manuscript for *Earth Surface Dynamics*, entitled "*Different coastal marsh sites reflect similar topographic conditions for bare patches and vegetation recovery*" [Paper #esurf-2020-56], and detailed list of our responses to the comments of the reviewer. We highly appreciated the comments made by the reviewer, as they enabled us to greatly improve the manuscript.

Below we give a step-by-step response to the comments. The original comments of the reviewer are copied below and shown in black. Our step-by-step replies are inserted and shown in blue. The line numbers that are mentioned refer to the line numbers in the revised manuscript with tracked changes.

Thank you very much for your continued consideration of this manuscript.

Looking forward to your reply.

Yours sincerely,

On behalf of all co-authors,

Chen Wang

**"Different coastal marsh sites reflect similar topographic conditions for bare patches and vegetation recovery" [Paper #esurf-2020-56]**

Chen Wang, Lennert Schepers, Matthew L. Kirwan, Enrica Belluco, Andrea D'Alpaos, Qiao Wang, Shoujing Yin, and Stijn Temmerman

**List of response to the comments**

Suggestions for revision or reasons for rejection (will be published if the paper is accepted for final publication)
The manuscript presents an interesting study of the occurrence of bare patches in three contrasting saltmarshes and aims to elucidate how topographic settings determine marsh destruction or revegetation. The introduction and methodology are very clear and the manuscript is generally well-written. However, I have some concerns about the mechanisms proposed in the discussion and the generalizations illustrated in figure 8. I moreover suggest to go through the text to correct some minor unclear formulations and corrections.

1) Main comments:

The manuscript claims that revegetation was studied for three sites but actually only one site showed revegetation. The other two sites were not investigated since there didn't occur revegetation as partly shown by previous studies. I suggest to rephrase more carefully in the abstract line 25 and the discussion line 345 that this is not a finding particular to this work but only a confirmation of previous studies.
This was adapted as suggested, by adding the following text (underlined here):
Line 26: "In line with previous studies, revegetation…."
Line 358-360: "No vegetation recovery was observed in the two sites with smaller tidal range and sediment input (Venice lagoon, Blackwater marshes), which is in line with previous studies in these two areas, showing progressive marsh die-off over longer (century) time scales (e.g. Schepers et al., 2017; Carniello et al., 2009 )."

The authors conclude that larger patches cause larger connecting channels but this is not clear from the findings. Alternatively, larger channels could also promote larger bare patches. I suggest to either rephrase the statement in the abstract line 21 and discussion line 358 or carry out an additional analysis tracking connected bare patch size and channel width through time to investigate what occurs first and determine which determines the other.
This was adapted as suggested, by adding the following text (underlined here):
Line 371-374: "First, concerning the positive relationship between bare patch size and connecting channel width, we are not certain about the direction of causal relationship (either larger bare patches causing wider connecting channels, or vice versa), but we may formulate certain hypotheses. This relationship may be due to…."
Line 21: "and that there is a positive relationship between the width of the connecting channels

and the size of the bare patches, in each of the three marsh sites."

➔ The latter formulation does not suggest the direction of causal relationship.

The discussion about the wave-induced resuspension is only valid for large microtidal marshes, such as Blackwater. I suggest to explore more on alternative hypotheses on tidal channel hydrodynamics (suggested references are in the detailed comments).
We followed this suggestion. The same suggestion comes back in the more detailed comments below, and there we explain how we adapted it in the manuscript.

Figure 8 generalizes the two findings from Blackwater and Saeftinghe in one figure, which should be separated: wave resuspension within bare patches and revegetation do not co-occur in the presented systems and therefore should be illustrated separately. I suggest to simplify the schematic model showing bed elevation and likelyhood that a certain feature occurs and possibly link to tidal range. Since it is not clear what is the main driver of bare patch formation (waves, SLR, SSC) I would refrain from generalizing the findings in a figure such as presented.
As the reviewer founds this figure confusing, and as it is only an attempt to present a conceptual, summarizing sketch, and as such it is not essential for the paper, we decided to leave this figure out.

2) Detailed comments:
3.1: I am missing a definition of how you define the difference between channel and connected bare patch/the boundary at which you define it as patch or channel?
An explanation was added:
Line 164-167: "The edge between a connected bare patch and the connecting channel was visually defined as where the channel planform shape (i.e. linearly shaped) in upstream (landward) direction widens into a bare patch (i.e. non-linear, more irregular shape), as shown in Figs. 1-3."

Line 165- 167: 'Field surveys only include selected 165 locations, but with greater vertical accuracy, especially for vegetated areas where LIDAR partially reflects on the vegetation canopy, and open water where LIDAR reflects on the water surface.'
It is not clear what this sentences means – do you mean accuracy is enhanced for the classes vegetated and open water?
We rewrote this sentence, so that it is clear what we mean:
L 170-173: "Field surveys only include selected locations, but field surveys of soil surface elevation had greater vertical accuracy than LIDAR surveys (see below), especially for vegetated areas where LIDAR partially reflects on the vegetation canopy, and open water where LIDAR reflects on the water surface."

Line 227: Please add a reference for the Mann-Whitney U test
Reference was added to: R Core Team, 2016. A Language and Environment for Statistical Computing. R, Vienna, Austria.

Line 233: Please add a reference after 'growth'

Reference was added to: Balke, T., Stock, M., Jensen, K., Bouma, T. J., and Kleyer, M., 2016, A global analysis of the seaward salt marsh extent: The importance of tidal range: Water Resources Research, v. 52, no. 5, p. 3775-3786.

5.1.1

You refer in this paragraph to the peak of the distribution but I am not certain how to interpret this value. Please explain in one sentence on what the distribution and the peak show.

This is explained:

Line 284-285: "The peaks of the elevation distribution (i.e. the mode of the elevation distribution)…"

Line 277-278: Please rephrase the sentence, to me it is not clear what this means - between all comparisons of two of the 3 variables?

This was reformulated as suggested:

Line 282-284: "The differences in elevation between the vegetated marshes, connected and unconnected bare patches were statistically significant between all comparisons of two of the three variables ($p < 0.001$ based on the Mann-Whitney test)."

5.1.3

I do not understand this paragraph from the figure. Where do I see the difference between connected and unconnected bare patches in Fig. 6? Where are the values (% of area) visible in the figure?

This is now explained in the text below the figure:

"Bare patches with connecting channel widths < 0.5 m are defined as unconnected bare patches in the text (see methods). The patch number proportion (%) is calculated as the number of bare patches in each class of bare patch size relative to the total number of bare patches for each category of channel width."

Line 347 key point (3): Please be careful with the phrasing: This last point does not emerge from the presented results but was already described in previous publications (as you mentioned in line 220 Schepers et al, 2017). Only the second sentence was shown in this work.

This was adapted in response to a previous similar remark by the reviewer above:

Line 358-360: "No vegetation recovery was observed in the two sites with smaller tidal range and sediment input (Venice lagoon, Blackwater marshes), which is in line with previous studies in these two areas, showing progressive marsh die-off over longer (century) time scales (e.g. Schepers et al. 2017; Carniello et al. 2009 )."

Line 336: What about the unconnected patches?

This was added:

Line 343-345: "Permanent bare areas are always connected to channels, and tend to be associated with wide channels, while unconnected bare patches always rapidly revegetated (i.e. within 4 to 6 years after their first appearance) (Fig. 7c)."

Line 350: 'high connectivity': Do you refer here to the width of the connecting channel in fig.7c?

Yes indeed. This is explained now explicitly:
Line 361-362: "…by wider channels connecting the bare patches to the channel network."

6.1
I think that the discussion focuses too much on the resuspension by waves but does not explore other optional hypotheses. For example, the time lag between incoming and outgoing tide can result in ebb-dominance in marshes and therefore lead to net sediment export through the channels (e.g. https://www.jstor.org/stable/25736162 or https://doi.org/10.1029/2019WR025942). I think this is a more elegant explanation for sediment export from the marsh and can explain the difference between Saeftinghe and Blackwater. This can furthermore be related to vegetation species, which are mentioned in the text but neglected in the discussion.
Moreover, I suggest to mention that waves are only relevant in microtidal marshes and negligible in the case of Saeftinghe, hence a possible explanation for the difference between the systems (see e.g. https://doi.org/10.1016/j.geomorph.2018.03.025).
We thank the reviewer for these useful suggestions, and included them in a reworked section of the discussion:
Line 379-390: Secondly, our finding that unconnected bare patches occur most frequently on higher elevations than connected bare patches, may be interpreted by a number of potential hypotheses. We expect that connected bare patches experience higher incoming and outgoing flood and ebb flow velocities as they are directly connected to the channels, while unconnected bare patches are surrounded by marsh vegetation, which is expected to obstruct and reduce flood and ebb flow velocities. Furthermore, the time-lag between incoming and outgoing tides can result in ebb-dominance in marshes (e.g. Friedrichs & Perry, 2001) and therefore may contribute to net sediment export from bare patches that are connected to the channel network. As such, stronger tidal currents, ebb-dominance and net sediment export may result in lower surface elevation of connected bare patches as compared to unconnected bare patches, where the surrounding vegetation may reduce flow velocities and facilitate the deposition of suspended sediments supplied during overmarsh tides. Such effects of tidal currents may be most pronounced in the study site with largest tidal range (Scheldt estuary), while additional effects of wind-waves on sediment transport have been reported to be important in the sites with intermediate and small tidal range (Venice lagoon, Blackwater marshes) (e.g. Stevenson et al. 1985; Fagherazzi et al. 2006).

I think that autocompaction and organic accretion by different vegetation species are also important phenomena to be mentioned.
Sorry, this suggestion is vague. It is not clear to us how this can contribute specifically to the discussion in 6.1.

Line 444-450: Please be more specific what you mean here. I am not sure what can be concluded from the above paragraph, what is the most likely explanation for large bare areas: SLR, tidal range or sediment supply? Or all these reasons?
This is further specified:
Line 474: "It is probably a combination of all these factors that may explain why…"

3) Figures:
Fig. 3:
Please add in the caption if the small study area is the only data considered in the paper - if yes, it is not clear to me why the blue patches are excluded but the yellow ones are considered.
Suggestion added in the figure caption:
"Data presented in this paper are for all bare patches in the small study area (both unconnected ones (in pink) and connected ones (in yellow)). In order to obtain a higher number of observations of connected bare patches, we also included connected bare patches in the larger study area (in yellow) but excluded unconnected bare patches (in blue)."

Fig.4 :
Please mention in figure caption again why there is no lidar data for bare patches at Blackwater and no field data for Saeftinghe
Suggestion added in the figure caption:
"Field data were added to LIDAR data for San Felice and Blackwater, because bare patches were partly covered there by water, which obstructs LIDAR sensing of the soil surface beneath the water surface; while in Saeftinghe all bare patches were drained at low tides and LIDAR is not obstructed here by water cover."

Fig. 6:
I got confused if both connected and unconnected patches were included. Only later I saw that the first class is unconnected channels. I think it would enhance clarity to add that in the figure by text but especially in the caption. Also, I miss the percentages mentioned in the text, maybe it is possible to highlight them.
Suggestion added in the figure caption:
"Bare patches with connecting channel widths < 0.5 m are defined as unconnected bare patches in the text (see methods). The patch number proportion (%) is calculated as the number of bare patches in each class of bare patch size relative to the total number of bare patches for each category of channel width."

Fig. 7:
Panel c is not clear to me: Why is 40% of pixels rapidly revegetated with a channel with of 0? I guess the two left data points are unconnected bare patches? I suggest to make that clear, maybe a scatter plot is more representative than a line plot since it is a limited amount of data and maybe separate connected/unconnected patches by a vertical line.
This is more clearly explained in the figure caption:
"The proportion (%) in panel (c) is calculated as the number of pixels in each class of channel width relative to the total number of pixels that are permanently bare patches (blue line) or rapidly revegated bare patches (red dashed line)."

4) Textual comments:

a) Some of the wording I am not familiar with, such as the terms 'overwash tide' and 'tidal frame'. Overwash is usually referred to as waves and I was not sure what the difference between tidal

range and tidal frame was.

We didn't use the term "overwash tide" but "overmarsh tide", these are high tides that submerge the complete marsh surface. This is explained now:

Line 130-131: "overmarsh tides (i.e. high tides that submerge the complete marsh surface)"

b) The use of the word 'connectivity' is arbitrary: do you mean connected/not connected or degree of connectivity through channel width? This should be defined in the introduction and possibly adjusted throughout the manuscript.

There were 6 places in the text where the word 'connectivity' was used. It was defined where it was used the first time as: "connectivity is defined here as the width of connecting channels"

c) You mix the use of the word 'feature' and 'category' for the different classes vegetated, bar patches etc. (e.g. fig.4 caption) . Please be consistent.

It was changed and the word 'category' was consistently used throughout the paper.

d) I suggest to revisit the punctuation in the manuscript, specifically the use of commas.

We revisit the punctuation as suggested.

4.1) Detailed comments:

All below suggestions were adapted:

Title: occurs = 'occur'

Line 19: 'distance from'

Line 151: 'tidal range' = 'tidal amplitude'

Line 212: 'method as for Saeftignhe' 'field elevation survey': remove 'elevation'

Line 231: 'the LIDAR data'

Line 364: 'on higher'= 'at higher'

Line 405: 'feedback'= 'feed back'

Line 406: 'remove second 'may' before 'contribute'

Line 478: 'indicative' is used twice in a row; 'recover from'

Caption figure 4: 'exact numbers' = 'total numbers'